# Boosting Parallel Algorithms in Linear Queries for Non-Monotone Submodular Maximization

## Abstract

In this work, we propose two efficient parallel algorithms, LinAst and LinAtg, that improve both the approximation ratio and query complexity of existing practical parallel algorithms for the non-monotone submodular maximization over the ground set of sized $n$ under a cardinality constraint $k$. Specifically, our algorithms keep the best adaptive complexity of $O(\log n)$ while significantly improving the approximation ratio from $1/6 - \epsilon$ to $0.193 - \epsilon$ and reducing the query complexity from $O(n \log(k))$ to $O(n)$. The key building block of our algorithms is LinAdapt, a constant approximation ratio within $O(\log n)$ sequence rounds and linear queries. LinAdapt can reduce the query complexity by providing $O(1)$ guesses of the optimal value. We further introduce the BoostAdapt algorithm returning a better ratio of $1/4 - \epsilon$ within $O(\log(n) \log(k))$ adaptive complexity and $O(n \log(k))$ query complexity. Our BoostAdapt works in a novel staggered greedy threshold framework that alternately constructs two disjoint sets in $O(\log k)$ sequential rounds. Besides theoretical analysis, the experiment results on validated benchmarks confirm the superiority of our algorithms in terms of solution quality, the number of required queries, and running time over cutting-edge algorithms.

## 1 Introduction

Maximizing a non-negative (not necessarily monotone) submodular set function under cardinality constraint is a fundamental and important problem that has a wide-range of applications in the fields of artificial intelligence and machine learning, such as data summarization (Kuhnle, 2021b; Lin & Bilmes, 2011; Chen & Kuhnle, 2023; Fahrbach et al., 2019a; Han et al., 2021; Mirzasoleiman et al., 2016), revenue maximization in social networks (Kuhnle, 2021b; Chen & Kuhnle, 2023), recommendation systems (Mirzasoleiman et al., 2016) and weight cut (Chen & Kuhnle, 2023; Kuhnle, 2021b). Given a finite ground set $V$ sized $n$, a submodular set function $f : 2^V \mapsto \mathbb{R}^+$ and a positive number $k$ (cardinality constraint), the submodular maximization under cardinality constraint (SMC) problem asks to find a set $S \subseteq V$ with $|S| \leq k$ so that $f(S)$ is maximized. The problem has received much attention in finding approximation solutions with theoretical bounds (Buchbinder et al., 2014; 2015; Kuhnle, 2019; Chen & Kuhnle, 2023; Gupta et al., 2010).

However, the problem must face exponentially increasing search space due to the explosion of input data. This motivates much effort to design efficient parallel algorithms that can power the parallel architectures of computer systems to obtain a good solution promptly (See Table 2 for an overview of parallelizable algorithms). In this context, the concept of *adaptive complexity* or *adaptivity* becomes an essential measure of the feasibility of any parallel algorithm. This concept is to evaluate the number of sequential rounds of an algorithm that can execute many independent polynomial oracle queries in parallel (Balkanski & Singer, 2018).

It is noted that improving adaptive complexity from $O(\log^2(n))$ to $O(\log n)$ dramatically reduces the number of sequential iterations, thereby significantly reducing the running time of the algorithms in practice (Fahrbach et al., 2019a; Ene & Nguyen, 2020; Amanatidis et al., 2021; Fahrbach et al., 2023; Kuhnle, 2021b; Chen & Kuhnle, 2024). Although the recent studies significantly reduce the adaptive complexity from $O(\log^2(n))$ to $O(\log n)$, they still face with many challenges, including:

(1) The high query complexity made the parallel algorithm in (Ene & Nguyen, 2020) impractical for real applications. (Chen & Kuhnle, 2024) demonstrated that (Ene & Nguyen, 2020)'s algorithm is of mostly theoretical interest because it needs $\Omega(nk^2 \log^2(n))$ queries to access the multi-linear extension of a submodular function and its gradient. Experimentally, its running on tiny instances (e.g., $n < 100$) is already prohibitive as it requires more than $10^9$ queries to the set function on an instance with $n = 87$.

(2) The solution qualities of the existing practical parallel algorithms (Fahrbach et al., 2019a; 2023; Kuhnle, 2021b; Chen & Kuhnle, 2024; Cui et al., 2023) are still low as they have a large gap in the approximation ratios compared to the best non-parallel algorithm for SMC (e.g. the ratio of $0.401$ in (Buchbinder & Feldman, 2024)).

This raises two interesting questions for us to solve SMC: **Q1: Can we improve the query complexity of a parallel algorithm? Q2: Can we improve the approximation ratio of a practical parallel algorithm?**

Table 1: Comparison of parallel algorithms for SMC, the best result(s) are bold.

| Reference | Ratio | Adaptivity | Query Complexity |
|---|---|---|---|
| (Gupta et al., 2010) | $1/6 - \epsilon$ | $O(nk)$ | $O(nk)$ |
| (Buchbinder et al., 2015) | $1/e - \epsilon \approx 0.367 - \epsilon$ | $O(k)$ | $O(n)$ |
| (Balkanski et al., 2018) | $1/(2e) - \epsilon \approx 0.183 - \epsilon$ | $O(\log^2(n))$ | $O(\mathsf{opt}n \log^2(n) \log(k))$ |
| (Chekuri & Quanrud, 2019) | $3 - 2\sqrt{2} - \epsilon \approx 0.171 - \epsilon$ | $O(\log^2(n))$ | $O(nk^4 \log^2(n))$ |
| (Ene & Nguyen, 2020) | $1/e - \epsilon \approx 0.367 - \epsilon$ | $O(\log n)$ | $\Omega(nk^2 \log^2(n))$ |
| (Fahrbach et al., 2023) (ANM) | $0.039 - \epsilon$ | $O(\log n)$ | $O(n \log(k))$ |
| (Amanatidis et al., 2021) | $0.172 - \epsilon$ | $O(\log n)$ | $O(nk \log(n) \log(k))$ |
| (Chen & Kuhnle, 2024) (AST) | $1/6 - \epsilon \approx 0.166 - \epsilon$ | $O(\log n)$ | $O(n \log(k))$ |
| (Chen & Kuhnle, 2024) (ATG) | $0.193 - \epsilon$ | $O(\log(n) \log(k))$ | $O(n \log(k))$ |
| (Cui et al., 2023)(ParSKP1) | $1/8 - \epsilon$ | $O(\log n)$ | $O(nk \log^2(n))$ |
| (Cui et al., 2023)(ParSKP2) | $1/4 - \epsilon$ | $O(\log(n) \log(k))$ | $O(nk \log^2(n))$ |
| LinAst (this work) | $1/6 - \epsilon$ | $\boldsymbol{O(\log n)}$ | $\boldsymbol{O(n)}$ |
| LinAtg (this work) | $\boldsymbol{0.193 - \epsilon}$ | $\boldsymbol{O(\log n)}$ | $\boldsymbol{O(n)}$ |
| BoostAdapt (this work) | $\boldsymbol{1/4 - \epsilon}$ | $\boldsymbol{O(\log(n) \log(k))}$ | $\boldsymbol{O(n \log(k))}$ |

**Our contributions and techniques.** In this work, we tackle the above questions with the following contributions:

- First, we introduce LinAdapt, the first constant approximation ratio $(1/(12 + O(\epsilon)))$ within near-optimal adaptivity $O(\log n)$ and linear query complexity $O(n)$, where $\epsilon > 0$ is a constant parameter. LinAdapt is used as a building block to reduce the query complexity of our latter algorithms. The key idea to obtain a such improvement lies in: **(1)** constructing two sets $S, S'$ in $O(\log n)$ sequential rounds with an interesting property: $f(X \cup S) \le O(1) \cdot f(S')$ with high probability for any set $X \subseteq V, |X| \le k$, and $S'$ with $|S'| \le k$ is the set of last elements added into $S$; **(2)** combining **(1)** appropriately with the unconstrained submodular maximization algorithm to get the desired ratio.

- Second, we introduce two algorithms LinAst and LinAtg which run $O(\log n)$ adaptivity and $O(n)$ query complexity and return the approximation ratios to $1/6 - \epsilon$ and $0.193 - \epsilon$, respectively. Therefore, our algorithm LinAtg improves the approximation ratio from $1/6 - \epsilon$ to $0.193 - \epsilon$ and significantly reduces the query complexity of the current best practical algorithm of (Chen & Kuhnle, 2024). Both LinAst and LinAtg use a common framework: first adapting LinAdapt as a subroutine to get the $O(1)$ number of guesses of the optimal, then using the adaptive simple threshold (Chen & Kuhnle, 2024) and iterated greedy framework (Gupta et al., 2010; Chen & Kuhnle, 2024) to get better theoretical bounds.

- Third, we further introduce BoostAdapt that provides a considerable approximation ratio of $1/4 - \epsilon$ in $O(\log(n) \log(k))$ adaptive complexity and $O(n \log(k))$ query complexity. Thus, BoostAdapt significantly improves the ratio of the ATG (Chen & Kuhnle, 2024)(the algorithm with same adaptive and query complexity) from $0.193 - \epsilon$ to $1/4 - \epsilon$ and also reduces the query complexity of ParSKP2 (Cui et al., 2023) (the algorithm with same approximation ratio and adaptive complexity) by a factor of $\Omega(k \log^2(n)/\log(k))$. BoostAdapt follows a novel *staggered threshold* framework: updating alternately two disjoint sets $X$ and $Y$ only in $O(\log k)$ iterations by adapting a threshold sampling. It must be noted that the staggered

threshold is different from the Iterated Greedy (Gupta et al., 2010), which uses two threshold greedy strategies to construct candidate solutions separately. Our algorithm also differs from the twin greedy strategy of (Han et al., 2020) and Interlaced Greedy of (Kuhnle, 2019), allowing simultaneously updating two disjoint sets in each iteration of the main loops.

- Finally, to show the consistency between theory and practice, we conducted several extensive experiments on two applications: Revenue Maximization and Maximum Weighted Cut. The results show that our algorithms not only produce the better solution quality and query complexity sharply but also provide comparative adaptivity to state-of-the-art algorithms.

**Paper Organization.** The rest of this work is structured as follows. Section 2 provides the literature review on the studied problem. Notations are presented in Section 3. Sections 4-6 introduce our algorithms and theoretical analysis. Experimental computation is provided in Section 7. Finally, Section 8 concludes this work.

## 2    RELATED WORKS

**Non-parallel Approximation Algorithms with Low Query Complexity.** The first popular approach to solving the SMC problem in practice is to design approximation algorithms with low query complexity. The greedy algorithm was an effective approach for submodular optimization problems. It sequentially selects elements with the largest marginal gain and explores the diminishing return property to get performance bounds. (Gupta et al., 2010) first developed an iterated greedy algorithm with $1/6$ ratio in $O(nk)$ queries for SMC. The work of (Lee et al., 2010) latter improve the ratio to $1/4 - \epsilon$ by developing a local search method but this wasted an expensive query complexity of $O(n^4 \log n)$. Significantly, the elegant random greedy in (Buchbinder et al., 2014) could provide the ratio of $1/e$ with $O(nk)$ query complexity. Instead of selecting an element with the best marginal gain as the naive greedy, it chose a uniformly random element from the set of $k$ elements with the largest marginal gain. (Buchbinder et al., 2015) then improved the query complexity to $O(n \log(1/\epsilon)/\epsilon^2)$ but still kept the same $1/e$ ratio. To the best of our knowledge, the best ratio for SMC was $0.385$ by (Buchbinder & Feldman, 2019). However, this work used the multi-linear extensions method and used a considerably high query complexity of $O(n^5)$ (Chen & Kuhnle, 2023). Besides, some efforts focused on devising deterministic approximations but did not improve the performance ratio or reduce the query complexity (Chen & Kuhnle, 2023; Kuhnle, 2019; Buchbinder & Feldman, 2018). Finally, it must be emphasized that the above works weren't well parallelized due to their high adaptive complexities of $\Omega(n)$.

**Parallel Algorithms with Low adaptive complexity.** Research on parallel algorithms was initiated by (Balkanski & Singer, 2018). In this seminal work, they introduced the concept of *adaptive complexity or adaptivity* that measuring the parallelizable of algorithm, showed a lower bound adaptive complexity of $O(\log(n)/\log(\log n))$ to achieve a constant ratio and devised an $1/3$-approximation algorithm in $O(\log n)$ adaptive rounds for the monotone SMC problem. Since then, many works focused on devising algorithms with low adaptivity with tighter ratio for SMC (Balkanski et al., 2019; Fahrbach et al., 2019b; Ene et al., 2019; Chen et al., 2021). The best parallel algorithm for monotone SMC had an optimal ratio of $1 - 1/e - \epsilon$ in $O(\log n)$ adaptive complexity and $O(n)$ queries was due to (Chen et al., 2021). However, their performance bounds heavily relied on the monotone property and did not hold for non-monotone.

For the non-monotone SMC, (Chekuri & Quanrud, 2019; Balkanski et al., 2018) first developed parallelizable algorithms with $3 - 2\sqrt{2} - \epsilon$ and $1/(2e) - \epsilon$ ratios, respectively; both took $O(\log^2 n)$ adaptivity. (Ene & Nguyen, 2020) improved the ratio to $(1/e - \epsilon)$ in nearly optimal adaptive complexity of $O(\log n)$. However, due to the high query complexity of $\Omega(nk^2 \log^2 n)$ to access the multilinear extension of a submodular function and its gradient. Therefore, the (Ene & Nguyen, 2020)' algorithm and algorithms based on multilinear extension methods in general are still impractical in some real applications (Kuhnle, 2021b; Fahrbach et al., 2019a; Chen & Kuhnle, 2024; Cui et al., 2023). To attack the above issue, (Fahrbach et al., 2019a) first aimed to reduce the query complexity to near-linear of $O(n \log k)$ and still kept the $O(\log n)$ adaptive complexity. However, their algorithm resulted in a small ratio, $0.039 - \epsilon$. In subsequent work, (Kuhnle, 2021b) tried to boost the ratio to $1/6 - \epsilon$ in $O(\log n)$ adaptive rounds with $O(n \log k)$ queries. Their algorithm further improved the ratio to $0.193 - \epsilon$ in $O(\log^2 n)$ adaptive rounds by exploring iterated greedy framework in (Gupta et al., 2010). It must be noted that the ratios of both (Fahrbach et al., 2019a) and (Kuhnle, 2021b)

may not hold because they adapt the threshold sampling routine in (Fahrbach et al., 2019b), which was pointed out disable with non-monotone functions (Chen & Kuhnle, 2024). Recently, (Chen & Kuhnle, 2024; Fahrbach et al., 2023) recovered the ratios of (Fahrbach et al., 2019a; Kuhnle, 2021b) by some threshold sampling routines with the respective analysis. More recent, (Cui et al., 2023) showed two algorithms with ratios of $1/8 - \epsilon$ and $1/4 - \epsilon$ with $O(\log n)$ and $O(\log^2 n)$ adaptive complexities, respectively. However, they took at least $\Omega(nk \log^2 n)$ query complexity.

## 3 PRELIMINARIES

Given a ground set $V$ of sized $n$, and the set function $f : 2^V \mapsto \mathbb{R}^+$ is submodularity iff it satisfies the diminishing return property, i.e., $f(A \cup \{e\}) - f(A) \geq f(B \cup \{e\}) - f(B)$ where $A \subseteq B$ and $e \notin B$. The marginal gain (or contribution gain) of a set $T$ to a set $S$ is defined as $f(T|S) = f(T \cup S) - f(S)$. For simplicity, we denote $f(\{e\}|A)$ by $f(e|A)$ and assume $f$ normalized, i.e., $f(\emptyset) = 0$. Given a positive number $k$ (cardinality constraint), the SMC problem is to determine $\arg \max_{S \subseteq V, |S| \leq k} f(S)$.

We define $[k] = \{1, 2, \ldots, k\}$ for any integer $k$. We denote an instance of SMC by a tuple $(f, V, k)$ and $O$ is an optimal solution with the optimal value opt $= f(O)$. In this work, we assume that there exists an oracle query that returns $f(S)$ when queried for the set $S$ with any $S \subseteq V$. The parallelization capacity of an algorithm is evaluated through the following definition.

**Definition 3.1** ((Balkanski & Singer, 2018)). *Given a value of oracle of $f$, the adaptivity or adaptive complexity of an algorithm is the minimum sequential number of rounds needed such that in each round the algorithm makes $O(\mathsf{poly}(n))$ independent queries to the evaluation oracle.*

We recap two sub-problems for solving SMC in parallel setting, Batch Selection with Threshold (BST) and unconstrained submodular maximization (USM). Given an instance $(f, V, B)$, a fixed threshold $\tau$ and $\epsilon > 0$ as inputs, BST asks to find a subset $S \subseteq V$ in $O(\log n)$ adaptive complexity satisfying two conditions: (1) $f(S) \geq (1 - \epsilon)\tau|S|$; (2) if $|S| < k$, $f(e|S) < \tau$ for any $e \notin S$.

Although several attempts exist to solve the BST, most only work with a monotone submodular function (Fahrbach et al., 2019b; Kazemi et al., 2019). In this work, we use the ThreshSeq algorithm in (Chen & Kuhnle, 2024) and its theoretical satisfaction to analyze our algorithm's performance in a nonmontone setting.

**Theorem 3.2** (Theorem 3 in (Chen & Kuhnle, 2024)). *Let $(f, V, k)$ be an instance of SMC. For any constants $\epsilon, \delta > 0$, the algorithm ThreshSeq outputs $A' \subseteq A \subseteq V$ such that the following properties hold: 1) The algorithm succeeds with probability at least $1 - \delta/n$. 2) There are $O(n/\epsilon)$ oracle queries in expectation and $O(\log(n/\delta)/\epsilon)$ adaptive complexity. 3) It holds $f(A') \geq (1 - \epsilon)\tau|A|$. If $|A| < k$, then $f(e|A) < \tau$ for all $e \in V$. 4) It also holds $f(A') \geq f(A)$ and $|A'| \geq (1 - \epsilon)|A|$.*

Regarding USM, which aims to find $\arg \max_{S \subseteq V} f(S)$, there exist several solving methods giving the constant approximation ratios such as the (Fahrbach et al., 2019a)'s algorithm (USM1) provides a ratio of $1/4 - \epsilon$ with probability at least $1 - \delta$ for the problem in $O(1)$ adaptive round and $O(\frac{1}{\epsilon} \log(\frac{1}{\delta}))$ queries and an essentially optimal algorithm of (Chen et al., 2019) (USM2) slightly improves the approximation ratio to $1/2 - \epsilon$ using $O(\log(1/\epsilon)/\epsilon)$ adaptive rounds and $O(n \log^3(1/\epsilon)/\epsilon^4)$ query complexity. Due to the space limitations, Pseudocodes for ThreshSeq and USM algorithms are given in the Appendix D.

## 4 LinAdapt: PARALLEL ALGORITHM WITH LINEAR QUERY COMPLEXITY

In this section, we introduce LinAdapt (Algorithm 1), the first constant ratio approximation algorithm within $O(\log n)$ adaptivity and $O(n)$ query complexity. LinAdapt receives an instance $(f, V, k)$, accuracy parameters $\epsilon, \delta > 0$, a constant $\alpha > 0$ and works in a novel algorithmic framework. LinAdapt first sequentially calls a subroutine LinBoundSet twice to find two candidates: one to get $A'$ over the ground set $V$ and the other to get $B'$ over a new ground set $V \setminus A'$. Note that $A$ and $B$ in LinAdapt are temporary sets that are

---

**Algorithm 1:** LinAdapt$(f, V, k, \alpha, \epsilon, \delta)$

---

1: **Input:** $f, V, k, \alpha, \epsilon, \delta$.
2: $A, A' \leftarrow \mathsf{LinBoundSet}(f, V, k, \epsilon, \delta/3)$
3: $B, B' \leftarrow \mathsf{LinBoundSet}(f, V \setminus A', k, \epsilon, \delta/3)$
4: $C \leftarrow \mathsf{USM1}(A', \epsilon, \delta/(3n))$
5: **return** $\arg \max_{S \in \{A', B', C\}} f(S)$

---

useful only for theoretical analysis. For any ground set $V$, LinBoundSet is a key subroutine with the following nice properties:

1) running in $O(\log(n)/\epsilon^2)$ and $O(n/\epsilon^3)$ query complexity.
2) returning two sets $A$ and $A'$ so that $f(A \cup X) \leq (8 + 2\alpha + \frac{2}{\alpha} + O(\epsilon))f(A')$, for any subset $X \subseteq V, |X| \leq k$ with high probability.

To easily follow the algorithm, we will latter provide a detailed analysis of LinBoundSet in Section 4.1. Finally, LinAdapt calls USM1 to find another candidate solution $C$ and returns the best among $A', B', C$.

Conceptually, our LinAdapt takes advantage of the properties LinBoundSet (by Theorem 4.4), and the fact that $f(O) \leq f(O \cup A) + f(O \cup B)$ to give the desired bound of solution. We state the LinAdapt's performance in Theorem 4.1.

**Theorem 4.1.** *For any input $(f, V, k)$, where $\epsilon, \delta \in (0, 1)$, LinAdapt runs in $O(\log(n/\delta)/\epsilon^2)$ adaptive complexity and $O(n/\epsilon^3)$ query complexity, returns a solution $S$ satisfying* $\mathsf{opt} \leq \left(8 + 2\alpha + \frac{2}{\alpha} + (\frac{16}{1-4\epsilon} + 2(1 + \frac{1}{\alpha})\frac{2(2-\epsilon)}{(1-\epsilon)(1-2\epsilon)})\epsilon\right)f(S)$ *with probability at least $1 - \delta/n$. If $\alpha = 1$, the algorithm achieves the best ratio of $\frac{1}{12+O(\epsilon)}$.*

*Proof.* Each subroutine terminates with failure probability $1/(3n)$. By the union bound of probabilities, the failure probability of the algorithm is bounded by $3 \cdot \frac{\delta}{3n} = \frac{\delta}{n}$. Let $a = 2 + \alpha + \frac{1}{\alpha} + (1 + \frac{1}{\alpha})\frac{2(2-\epsilon)}{(1-\epsilon)(1-2\epsilon)}\epsilon$. If the algorithm terminates successfully, we have:

$$f(O) \leq f(O \cup A) + f(O \cup B) \tag{1}$$
$$\leq f(O \cup A) + f((O \cup B) \setminus A') + f((O \cup B) \cap A') \tag{2}$$
$$= f(O \cup A) + f((O \setminus A') \cup B) + f(O \cap A') \tag{3}$$
$$\leq a(f(A') + f(B')) + \frac{4}{1-4\epsilon}f(C) \tag{4}$$
$$\leq (2a + \frac{4}{1-4\epsilon})f(S) = \left[8 + 2\alpha + \frac{2}{\alpha} + \left(\frac{16}{1-4\epsilon} + 2(1+\frac{1}{\alpha})\frac{2(2-\epsilon)}{(1-\epsilon)(1-2\epsilon)}\right)\epsilon\right]f(S)$$

where inequalities equation 1 is due to the submodularity of $f$; inequalities equation 2 is due to the submodularity of $f$ and the fact that $A \cap B = \emptyset$, and inequality equation 4 is due to Theorem 4.4 and USM1's performance. For the complexities, the algorithm calls USM1 and LinAdapt twice so its adaptive complexity is $2 \cdot O(\frac{\log n}{\epsilon^2}) + 1 = O(\frac{\log n}{\epsilon^2})$ and query complexity is $2 \cdot O(\frac{n}{\epsilon^3}) + O(\frac{1}{\epsilon}\log(\frac{n}{\epsilon})) = O(\frac{n}{\epsilon^3})$. □

### 4.1 A KEY SUBROUTINE: LinBoundSet

LinBoundSet (Algorithm 2) receives an instance $(f, V, k)$, accuracy parameters $\epsilon, \delta > 0$ and a positive constant $\alpha$. It first initiates a set $S$ that contains an element with maximal value $e_{max}$ and then operates in $\ell = O(\log n)$ iterations of the main loop (Lines 3-21). Denote $S_j$ as $S$ after iteration $j$. At each iteration of main loop, it generates a random permutation of $V$ (Line 5) and divides $V$ into blocks $T'_{\lambda_i}, \forall \lambda_i \in \Lambda$. We define an element $v_t \in T'_{\lambda_i}$ that is **good** if $f(v_t|\{v_1, \ldots, v_{t-1}\}) \geq \alpha M_{\lambda_i}/k$, where $M_{\lambda_i} = \max_{j=0\ldots i} f(S \cup T_{\lambda_j}), i \in [m]$; a block $T'_{\lambda_i}$ is **good** if it has at least $(1-\epsilon)\alpha|T'_{\lambda_i}|$ good elements.

At a high level, at iteration $j$, the algorithm selects a segment of elements $T_{\lambda^*}$ and its subset $T^j$ (Line 21) from set $T$, a random permutation of $V$ satisfying three following requirements with high probability: (1) $|T^j| \geq (1-\epsilon)|T_{\lambda^*}|$; (2) $f(T^j|S) \geq (1 - O(\epsilon))\alpha|T_{\lambda^*}|f(S)/k$; and (3) remove elements in $V$ with the marginal gain is less than $\alpha f(S)/k$, i.e., $f(e|S) < \alpha f(S)/k$ for all $e \in V \setminus S$.

To achieve the requirements, the algorithm first removes every element whose marginal gain is less than $\alpha f(S)/k$ (Line 3); later, it selects $T_{\lambda^*}$, a consecutive sequence of elements $T$ that contains at least $(1-\epsilon)$-fraction number of blocks that are good by finding $\lambda^*$ in Line 19. Finally, it selects new block $T''_{\lambda_i}$ that only contains elements with non-negative marginal gain from $T'_{\lambda_i}$ (Line 20) and

---

**Algorithm 2:** LinBoundSet$(f, V, k, \alpha, \epsilon, \delta)$

---

1: **Input:** $f, V, k, \alpha, \epsilon, \delta$
2: $e_{max} \leftarrow \max_{e \in V} f(e), S \leftarrow \{e_{max}\}, \beta \leftarrow \epsilon \log((1 - e^{\frac{-\epsilon}{2}})/8)/16, \ell \leftarrow \lceil (4 + \frac{4}{\beta\epsilon}) \log(\frac{n}{\delta}) \rceil$
3: **for** $j \leftarrow 1$ to $\ell$ **do**
4:      $V \leftarrow \{x \in V : f(x|S) \geq \alpha f(S)/k\}$
5:      **If** $V = \emptyset$ **then** break;
6:      $V = \{v_1, v_2, \ldots, v_{|V|}\} \leftarrow$ rand-permutation$(V)$
7:      $\Lambda_1 \leftarrow \{\lfloor (1+\epsilon)^l \rfloor : 1 \leq \lfloor (1+\epsilon)^l \rfloor \leq k, l \in \mathbb{N}\}$
8:      $\Lambda_2 \leftarrow \{\lfloor k + l\epsilon k \rfloor : \lfloor k + l\epsilon k \rfloor \leq |V|, l \in \mathbb{N}\} \cup \{|V|\}$
9:      $\Lambda = \{\lambda_1, \ldots, \lambda_m\} \leftarrow \Lambda_1 \cup \Lambda_2, T_i = \{v_1, v_2, \ldots, v_i\}, T'_{\lambda_i} \leftarrow T_{\lambda_i} \setminus T_{\lambda_{i-1}}$
10:      Calculate $M_{\lambda_i} \leftarrow \max_{\lambda_j \leq \lambda_i} f(S \cup T_{\lambda_j})$ (in parallel)
11:      $b[v_i] \leftarrow$ **none**, $\forall v_i \in V; B[\lambda_i] \leftarrow$ **false**, $\forall \lambda_i \in \Lambda$
12:      **for** $\lambda_i \in \Lambda$ (in parallel) **do**
13:          **for** $v_l \in T'_{\lambda_i}$ (in parallel) **do**
14:              **if** $f(v_l|S \cup T_{l-1}) \geq (1-\epsilon)\alpha M_{\lambda_{i-1}}/k$ **then** $b[v_l] \leftarrow$ **true**
15:              **else if** $f(v_l|S \cup T_{l-1}) < 0$ **then** $b[v_l] \leftarrow$ **false**
16:          **if** $|\{v \in T'_{\lambda_i} : b[v] = $ **true**$\}| \geq (1-\epsilon)|T'_{\lambda_i}|$ **then** $B[\lambda_i] \leftarrow$ **true**
17:      $\lambda_1^* \leftarrow \max\{\lambda_i \in \Lambda, \lambda_i < k : B[\lambda_i] = $ **false**$, B[\lambda_1] = B[\lambda_2] = \ldots = B[\lambda_{i-1}] = $ **true**$\}$
18:      $\lambda_2^* \leftarrow \max\{\lambda_i \in \Lambda, \lambda_i \geq k : \exists u \geq 1 \; s.t. |\bigcup_{j=u}^{i-1} T'_{\lambda_j}| \geq k$ and $(B[\lambda_u] = B[\lambda_2] = \ldots =$
       $B[\lambda_{i-1}] = $ **true**$\}$
19:      $\lambda^* \leftarrow \max\{\lambda_1^*, \lambda_2^*\}$
20:      Define new blocks: $T''_{\lambda_i} \leftarrow \{v \in T'_{\lambda_i} : b[v] \neq $ **false**$\}$
21:      $T^j \leftarrow \bigcup_{\lambda_i \leq \lambda^*} T''_{\lambda_i}, S \leftarrow S \cup T^j$
22: **If** $|V| > 0$, **then return** $failure$
23: $S' \leftarrow$ last blocks added into $S$ with the size at most $k$
24: **return** $S, S'$

---

selects all new blocks from $T_{\lambda^*}$ (Line 21). At the end of each iteration, it adds $T_j$ into $S$ and the main loop ends after $\ell$ iterations or $V$ is empty. Finally, the algorithm returns $S'$ as the union of last $T''_{\lambda_i}$ blocks added to $S$ so that the size of $S'$ does not exceed $k$ (Line 23).

The analysis LinBoundSet's performance works in following process. We first focus on the bridge between $S$ and $S'$ in Lemma 4.2.

**Lemma 4.2.** *If* LinBoundSet *ends successfully, then* $f(S) \leq \left(1 + \frac{1+\epsilon}{(1-\epsilon)(1-2\epsilon)\alpha}\right) f(S')$.

We define an *iteration $j$ succeeds* if the algorithm successfully filters out more than $\beta\epsilon$-fraction of $V$ at the next iteration. Otherwise, the iteration *$j$ fails*. The following Lemma provides a bound of probability for the event that iteration $j$ fails.

**Lemma 4.3.** $\Pr[iteration \; j \; fails\,] \leq 1/2$.

Combine Lemma 4.2 and Lemma 4.3, we state the performance of LinBoundSet in Theorem 4.4.

**Theorem 4.4.** *For any* $\epsilon, \delta \in (0, 1)$, *the algorithm* LinBoundSet *runs in* $O(\log(n/\delta)/\epsilon^2)$, *returns two sets $S$ and $S'$ such that the following holds unconditionally: (1) The algorithm succeeds with a probability of at least* $1 - \delta/n$; *(2) The algorithm takes* $O(n/\epsilon^3)$ *query complexity in expectation; (3) If the algorithm succeeds, it returns $S, S'$ satisfying: for any subset $X \subseteq V, |X| \leq k$, we have*

$$f(S \cup X) \leq \left(2 + \alpha + \frac{1}{\alpha} + (1 + \frac{1}{\alpha})\frac{2(2-\epsilon)}{(1-\epsilon)(1-2\epsilon)}\epsilon\right) f(S').$$

Due to the space limit, the proofs of Lemma 4.2, Lemma 4.3, and Theorem 4.4 are in Appendix E.

## 5    IMPROVED RATIO ALGORITHMS WITH LINEAR QUERY COMPLEXITY

This section introduces two linear query and near-optimal adaptivity algorithms: LinAtg and LinAst. Both share a common framework in which LinAdapt plays a central role and provides $O(1)$ number of guesses of the optimal solution. Firstly, they adapt LinAdapt with

$\alpha = 1, \delta = 1/3$ to give a candidate solution $S_0$ (Line 2 of Algorithm 3 and Algorithm 4) with opt $\in [f(S_0), (12 + O(\epsilon))f(S_0)]$. It thus provides $O(\frac{1}{\epsilon} \log(\frac{1}{\epsilon}))$ guesses of opt. They then use the Adaptive Simple Threshold (AST) framework (Chen & Kuhnle, 2024) and the iterated greedy algorithm framework (Gupta et al., 2010; Chen & Kuhnle, 2024) to boost the approximation ratio.

In particular, LinAst constructs $O(\frac{1}{\epsilon} \log(\frac{1}{\epsilon}))$ solutions in the main loop with one adaptive round (Lines 4-9, Algorithm 3). For each iteration, the algorithm calls ThreshSeq twice sequentially to get candidate solutions $A_i'$, $B_i'$ with a threshold $\tau_i$ related to the guess of optimal value.

Then, LinAst finds another solution by using USM2 algorithm of (Chen et al., 2019) for USM over the ground set $A_i'$. Finally, it returns the best among the obtained solutions (Line 10, Algorithm 3).

LinAtg works in a different way. It sequentially constructs two pairs of disjoint sets $(A, B)$ and $(A', B')$ in two main loops which contain at most $\ell = O(\frac{1}{\epsilon} \log(\frac{1}{\epsilon}))$ iterations (Lines 5-14, Algorithm 4). At each iteration, the algorithm calls ThreshSeq over appropriate ground sets to select two batches of elements with high marginal gain $(S, S')$ and adds them into $(A, A')$ (or $(B, B')$), respectively. It then selects a candidate solution by adapting the USM2 algorithm of (Chen et al., 2019) over the ground set $A'$. Finally, LinAtg returns the best one among the obtained feasible solutions (Line 16, Algorithm 4).

We provide the performance of LinAtg and LinAst in Theorems 5.2, 5.1. Their proofs are provided in Appendix E.

**Theorem 5.1.** *For any input $(f, V, k, \epsilon)$, where $\epsilon \in (0, 1)$, LinAst runs in $O(\log(n)/\epsilon^2)$ adaptive complexity and $O(n/\epsilon^3)$ query complexity and returns an approximation ratio of $1/6 - \epsilon$ in expectation with probability of at least $1 - 1/n$.*

**Theorem 5.2.** *For any input $(f, V, k, \epsilon)$, where $\epsilon \in (0, 1)$, LinAtg runs in $O(\log(1/\epsilon) \log(n)/\epsilon^2)$ adaptive complexity and $O(n/\epsilon^3)$ query complexity, returns an approximation ratio of $0.193 - \epsilon$ in expectation with probability of at least $1 - 1/n$.*

---

**Algorithm 3:** LinAst

1: **Input:** $f, V, k, \epsilon$.
2: $S_0 \leftarrow \mathsf{LinAdapt}(f, V, k, 1, \epsilon, 1/3)$, $a \leftarrow 12 + (\frac{16}{1-4\epsilon} + \frac{8(2-\epsilon)}{(1-\epsilon)(1-2\epsilon)})\epsilon$
3: $c \leftarrow 6 + \epsilon, \ell \leftarrow \lceil \log_{1-\epsilon}(\frac{1}{a}) \rceil + 1, \delta \leftarrow 1/3$, $M \leftarrow af(S_0)/(ck)$
4: **for** $i \leftarrow 1$ **to** $\ell$ (in paralell) **do**
5:     $\tau_i \leftarrow M(1 - \epsilon)^i$
6:     $A_i, A_i' \leftarrow \mathsf{ThreshSeq}(f(\cdot), V, k, \tau_i, \epsilon, \delta)$
7:     $B_i, B_i' \leftarrow \mathsf{ThreshSeq}(f(\cdot), V \setminus A_i, k, \tau_i, \epsilon, \delta)$
8:     $A_i'' \leftarrow \mathsf{USM2}(A_i', \epsilon')$
9:     $C_i \leftarrow \arg\max\{f(A_i'), f(B_i'), f(A_i'')\}$
10: $S \leftarrow \arg\max_{X \in \{C_i\}_{i=1}^{\ell} \cup \{S_0\}} f(X)$
11: **return** $S$

---

**Algorithm 4:** LinAtg

1: **Input:** $f, V, k, \epsilon$.
2: $S_0 \leftarrow \mathsf{LinAdapt}(f, V, k, 1, \epsilon, 1/3)$, $a \leftarrow 12 + (\frac{16}{1-4\epsilon} + \frac{8(2-\epsilon)}{(1-\epsilon)(1-2\epsilon)})\epsilon$
3: $c \leftarrow 8/\epsilon, \epsilon' \leftarrow (1 - 1/e)\epsilon/8, \ell \leftarrow \lceil \log_{1-\epsilon'}(\frac{1}{ac}) \rceil + 1, \delta \leftarrow 1/(3\ell)$, $M \leftarrow af(S_0)/k$
4: $A \leftarrow \emptyset, A' \leftarrow \emptyset, B \leftarrow \emptyset, B' \leftarrow \emptyset$
5: **for** $i \leftarrow 1$ **to** $\ell$ **do**
6:     $\tau \leftarrow M(1 - \epsilon')^{i-1}$
7:     $A, S' \leftarrow$ $\mathsf{ThreshSeq}(f(A \cup \cdot), A, k - |A|, V, \epsilon', \delta, \tau)$
8:     $A \leftarrow A \cup S, A' \leftarrow A' \cup S'$
9:     **If** $|A| = k$ **then** break
10: **for** $i \leftarrow 1$ **to** $\ell$ **do**
11:     $\tau \leftarrow M(1 - \epsilon')^{i-1}$
12:     $S, S' \leftarrow$ $\mathsf{ThreshSeq}(f(B \cup \cdot), V \setminus A, k - |B|, \epsilon', \delta, \tau)$
13:     $B \leftarrow B \cup S, B' \leftarrow B' \cup S'$
14:     **If** $|B| = k$ **then** break
15: $A'' \leftarrow \mathsf{USM2}(A', \epsilon')$
16: $C \leftarrow \arg\max\{f(A'), f(B'), f(A''), f(S_0)\}$
17: **return** $C$

---

## 6 BoostAdapt: BOOST PARALLEL ALGORITHM WITH NEAR-LINEAR QUERY COMPLEXITY

This section introduces our last algorithm, BoostAdapt (Algorithm 5), which further improves the ratio to $(1/4 - \epsilon)$ in $O(\log(n) \log(k))$ adaptivity and $O(n \log(k))$ query complexity.

Different from LinAtg and LinAst, after reusing LinAdapt's solution (Line 2), BoostAdapt operates in a novel staggered strategy that consists of a main loop with $O(\log(k/\epsilon)/\epsilon)$ iterations (Lines 5-11). It initiates two disjoint sets $X, Y$ and their subsets $X', Y'$. At each iteration, it only updates alternatively partial solutions, either $X, X'$ or $Y, Y'$, by calling ThreshSeq (Lines 8, 10). The algorithm then

carefully selects candidates $X''$ and $Y''$ that consist of $\min\{k, |X'|\}$ and $\min\{k, |Y'|\}$ from $X'$ and $Y'$, respectively (Lines 13-14). Finally, it returns the best among candidates without violating the cardinality constraint (Line 15). At a high level, our BoostAdapt algorithm do not adapt USM

---

**Algorithm 5:** BoostAdapt

1: **Input:** $f, V, k, \epsilon$.
2: $S_0 \leftarrow \mathsf{LinAdapt}(f, V, k, 1, \epsilon, 1/3), a \leftarrow 12 + (\frac{16}{1-4\epsilon} + \frac{8(2-\epsilon)}{(1-\epsilon)(1-2\epsilon)})\epsilon$
3: $M \leftarrow af(S_0), \Delta \leftarrow \lceil \log_{\frac{1}{1-\epsilon}}(\frac{ak}{4\epsilon}) \rceil + 1, \delta \leftarrow 1/(3\Delta)$
4: $X, X', Y, Y' \leftarrow \emptyset, k' \leftarrow \max\{i \in \mathbb{N} : (1-\epsilon)i \leq k\}, \tau \leftarrow \frac{k'M}{4k}$
5: **for** $i \leftarrow 1$ to $\Delta$ **do**
6:   **if** $i$ is **odd then**
7:     $\tau_X \leftarrow \tau(1-\epsilon)^i, (A_i, A'_i) \leftarrow \mathsf{ThreshSeq}(f(X \cup \cdot), V \setminus (X \cup Y), k' - |X|, \epsilon, \delta, \tau_X)$
8:     $X \leftarrow X \cup A_i, X' \leftarrow X' \cup A'_i$
9:   **else**
10:     $\tau_Y \leftarrow \tau(1-\epsilon)^i, (B_i, B'_i) \leftarrow \mathsf{ThreshSeq}(f(Y \cup \cdot), V \setminus (X \cup Y), k' - |Y|, \epsilon, \delta, \tau_Y)$
11:     $Y \leftarrow Y \cup B_i, Y' \leftarrow Y' \cup B'_i$
12: Define $X' = \{x'_1, x'_2, \ldots, x'_{|X'|}\}, Y' = \{y'_1, y'_2, \ldots, y'_{|Y'|}\}$ contain elements in the order selected.
13: $X'' \leftarrow$ set of $\min\{k, |X'|\}$ elements $x'_i \in X'$ with largest marginal gain $f(x'_i|X'^{<x'_i})$
14: $Y'' \leftarrow$ set of $\min\{k, |Y'|\}$ elements $y'_i \in Y'$ with largest marginal gains $f(y'_i|Y'^{<y'_i})$
15: $S \leftarrow \arg\max_{Z \in \{X', Y', X'', Y'', S_0\}, |Z| \leq k} f(Z)$
16: **return** $S$

---

algorithms (may make the approximation ratio worse) and operates differently from Iterated Greedy (Gupta et al., 2010), which inspired AdaptiveThresholdGreedy of (Kuhnle, 2021b; Chen & Kuhnle, 2024). BoostAdapt is an elegant combination between threshold greedy (Badanidiyuru & Vondrák, 2014) and twin greedy (Han et al., 2020) via using ThreshSeq procedures alternatively to update two disjoint solutions $X$ and $Y$. Interestingly, we found that $X$ and $Y$ can support each other to bound elements in the optimal solution. Besides, by carefully analyzing the role of the set $X$ after the first iteration (i.e., $X_1$), we can give a better ratio than previous algorithms.

Before analyzing the BoostAdapt's performance, we provide some useful notations as follows: $X^i$, $Y^i$ are the sets of first $i$ elements added into $X$ and $Y$, respectively; $X_i$ and $Y_i$ are $X$ and $Y$ after the iteration $i$ of the main loop and $X_0 = Y_0 = \emptyset$; $X'_i$ and $Y'_i$ are defined analogously. Define $A_i^+$ (res. $B_i^+$) as the set of elements in $A_i$ (res. $B_i$) having the marginal gain at least $\tau_X$ (res. $\tau_Y$) at iteration $i$, $X_i^+ = \cup_{j=1}^i A_j^+, Y_i^+ = \cup_{j=1}^i B_j^+$ and $X^+ = X_\Delta^+, Y^+ = Y_\Delta^+$. For an element $e \in X \cup Y$, we denote by $X^{<e}, Y^{<e}$ as $X, Y$ right before $e$ is selected into $X$ or $Y$, respectively.

By carefully analyzing the relationship between $X$ and $Y$ through thresholds, we further provide connection between $X$ and $Y$ after each iteration in Lemma 6.1.

**Lemma 6.1.** *If* BoostAdapt *succeeds and $X_1 = \emptyset$, after iteration $i$ the following properties hold:*
*(a) If $i$ is odd and $|X_i| < k'$, for any set $C \subseteq Y_{i-1}^+$, we have $\sum_{x \in C} f(x|X_i) \leq \sum_{x \in C} \frac{f(e|Y^{<e})}{1-\epsilon}$.*
*(b) If $i$ is even and $|Y_i| < k'$, for any set $D \subseteq X_{i-1}^+$, we have $\sum_{x \in D} f(x|Y_i) \leq \sum_{x \in D} \frac{f(e|X^{<e})}{1-\epsilon}$.*

Using Lemma 6.1 and the fact that if $|T| = k$, then $|T \setminus O| \geq |O \setminus T|$, where $T \in \{X^+, Y^+\}$, we can bound of the optimal solution in some cases related to the size of $X$ and $Y$ in Lemma 6.2.

**Lemma 6.2.** *If* BoostAdapt *succeeds and $X_1 = \emptyset$, at the end* BoostAdapt *we have*
*a) If $|X| < k'$ and $|Y| < k'$, then $f(O) < \frac{(4-2\epsilon)f(S)}{(1-\epsilon)^2(1-2\epsilon)}$.*
*b) If there exists $T \in \{X, Y\}$ such that $|T| = k'$, then $f(S) \geq (1-\epsilon)^4 \frac{\mathsf{opt}}{4}$.*

Finally, putting them together, we give the tighter ratio in Theorem 6.3.

**Theorem 6.3.** *For any $\epsilon \in (0, 1/4)$ the algorithm runs in $O(\log(n/\epsilon)\log(k/\epsilon)/\epsilon^2)$ adaptive complexity, $O(n\log(k/\epsilon)/\epsilon^2)$ query complexity in expectation. The algorithm succeeds with a probability of at least $1 - 1/n$. If the algorithm succeeds, it returns an approximation ratio of $1/4 - \epsilon$.*

*Proof.* Due to the space limit, we put proof for successful probability and complexities in Appendix G. If the algorithm succeeds, we now prove the ratio by considering the following cases: (1) If $X_1 \neq \emptyset$ or $Y_2 \neq \emptyset$, we have $f(S) \geq \max\{f(X_1), f(Y_2)\} \geq \frac{(1-\epsilon)^2 k'M}{4k} \geq (1-\epsilon)^2 \frac{\mathsf{opt}}{4} > (\frac{1}{4} - \epsilon)\mathsf{opt}$. (2) If $X_1$ and $Y_2$ are both empty. If $|X| < k'$ and $|Y| < k'$, from Lemma 6.2 and $\epsilon < 1/4$, we have

$$f(S) > \frac{(1-\epsilon)^2(1-2\epsilon)}{4-2\epsilon}\mathsf{opt} > (\frac{1}{4} - \frac{7\epsilon}{4(2-\epsilon)})\mathsf{opt} \geq (\frac{1}{4} - \epsilon)\mathsf{opt}. \tag{5}$$

If there exists $T \in \{X, Y\}$ such that $|T| = k$, then $f(S) \geq (1-\epsilon)^4 \frac{\mathsf{opt}}{4} > (\frac{1}{4} - \epsilon)\mathsf{opt}$. This completes the proof. $\square$

# 7 EMPIRICAL STUDY

In this section, we experiment our LinAst, LinAtg and BoostAdapt by comparing to state-of-the-art for non-monotone SMC including IteratedGreedy (IG) (Gupta et al., 2010), FastRandomGreedy (FRG) (Buchbinder et al., 2015), AdaptiveNonmonotoneMax (ANM) (Fahrbach et al., 2019a), AdaptiveSimpleThreshold (AST) (Chen & Kuhnle, 2024), and AdaptiveThresholdGreedy (ATG) (Chen & Kuhnle, 2024). The comparison is about **four metrics**: object values, adaptive complexity, number of queries, and running time. We experimented with two well-known applications: Revenue Maximization (RM) and Maximum Cut (MC) (Chen & Kuhnle, 2024; Kuhnle, 2019; Amanatidis et al., 2020).

**Dataset and Setting.** For all algorithms, we set $\epsilon = 0.1$. Our algorithms were set $\alpha = 4.0$. Algorithms were run 20 times and averaged the results. We used ca-Astro ($n = 18,772, m = 198,110$) for RM and utilized web-Google ($n = 875,713, m = 5,105,039$), ca-GrQc ($n = 5,242, m = 14,496$), and Barabasi-Albert ($n = 968, m = 5,708$) for MC. These standard datasets were sourced from SNAP[1]. Appendix H gives a more detailed setting.

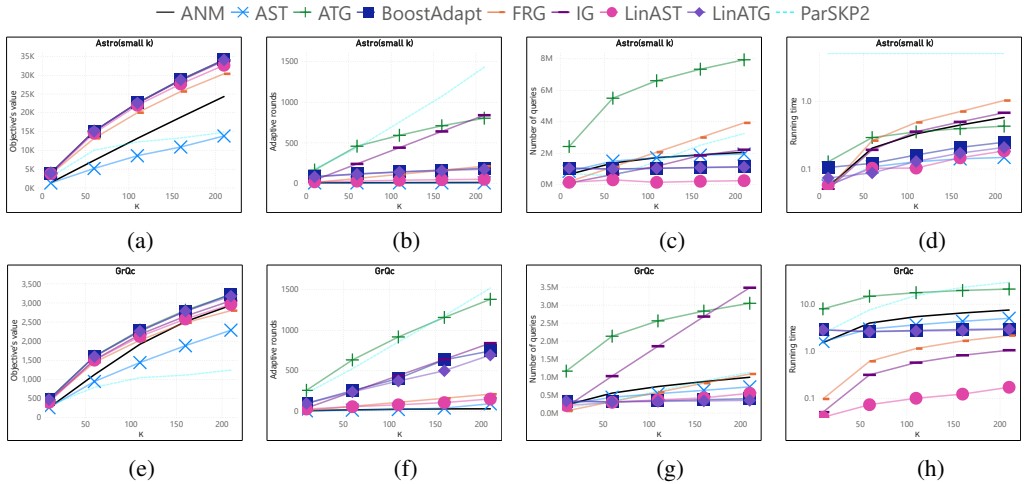

(a)   (b)   (c)   (d)

(e)   (f)   (g)   (h)

Figure 1: Performance of algorithms on Revenue Maximization (a-d) and Maximum Cut (e-h).

**Overview of Results.** Figure 1 displays the performance of compared algorithms on the ca-Astro and ca-GrQC datasets for RM and MC, respectively. Additional results are presented in Appendix H.

**Objective value:** Figures 1(a)(e) show the objective values of algorithms. It can be observed that the lines of BoostAdapt consistently reach the highest points with every $k$. Equivalent to BoostAdapt is LinAtg, IG and AST. Both LinAst and FRG mark a little lower than BoostAdapt, followed by ANM, AST, and ParSKP Finally, our algorithms significantly improve the solution quality.

**Adaptive rounds:** In the MC application (Figure 1(f)), the adaptive rounds result three distinct groups. The group with low adaptivity includes ANM, AST, LinAst, and FRG while the medium

---

[1]https://snap.stanford.edu/data/

group includes IG, LinAtg, and BoostAdapt. The high group is the remaining. ParSKP and ATG algorithms waste the highest number of adaptive rounds while ANM hits the lowest points. With the RM application (Figure 1(b)), ParSKP, ATG and IG give high adaptivity. When $k$ increases, their number of adaptive rounds can quickly grow to 4-10 times larger than the others. LinAtg slightly differs from the lowest ANM while LinAst and BoostAdapt are higher than ANM, but they do not exceed 200.

**Number of queries:** In both RM and MC (Figures 1(c) and (g)), our algorithms almost always minimize the number of queries. In RM (Figure 1(c)), LinAst has the lowest number of queries, followed by LinAtg and BoostAdapt algorithms. Meanwhile, AST is almost close to ANM and slightly higher than BoostAdapt. The remaining ATG, FRG, ParSKP, and IG belong to the group with many queries. The ATG algorithm wastes the highest number of queries. For MC (Figure 1(g)), LinAtg has the lowest number of queries, followed by BoostAdapt and LinAst. The algorithms AST, ParSKP, FRG, and ANM are again at an average level. while ATG and IG typically require the highest number of queries, about 5 times greater than the lowest line, LinAtg. On the whole, all our algorithms save queries more than the others. This is consistent with (nearly) linear query complexities of our algorithms.

**Time taken:** In MC (Figure 1(h)), LinAst wastes the lowest time, followed by IG and FRG. While BoostAdapt and LinAtg have the same running time, which is higher than IG and FRG but lower than AST, ParSKP and ANM. The gap among these considerably reduces, especially when $k$ increases, except ParSKP. Meanwhile, the time consumption of ParSKP steadily grows up along with $k$'s growth. In RM (Figure 1(d)), LinAst and AST have the lowest running time, followed by the group of LinAtg, BoostAdapt, ATG, but with a small gap. the rest uses more time than the others, especially ParSKP, which wastes the highest, about 10-20 times higher than the others.

The above metrics show that our algorithms outperform the others when keeping the best quality solutions, wasting the lowest query numbers within acceptable low adaptive rounds.

## 8 CONCLUSIONS

Motivated by the big data challenge for non-monotone submodular maximization under cardinality constraint, in this work, we focus on parallel approximation algorithms based on the concept of adaptive complexity. In particular, we proposed efficient parallel algorithms that significantly improve both approximation ratio and query complexity but keep the near-optimal adaptive complexity of $O(\log n)$. Our algorithm also expresses superior solution quality and computation complexity compared to state-of-the-art algorithms. However, there is still a weakness in our contribution, which is about the approximation factor. It leads to an opening question: how to reduce the gap between ours and the best ratio for $O(\log n)$ adaptive complexity in (Ene & Nguyen, 2020)?

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

## A  ORGANIZATION OF THE APPENDIX

- Section B presents simplified version of BoostAdapt algorithms and discussion.
- Section C presents some essential Probability Lemmas and Concentration Bounds used in this work.
- Section D presents algorithms missed in Section 3.
- Section E presents the proofs missed of Section 4.
- Section G presents the proofs missed of Section 6.
- Section H presents the additional experimental details and results of Section 7.
- Section I provides additional related work for monotone SMC.

## B  SIMPLIFIED VERSION OF BoostAdapt ALGORITHM

To easily follow the idea of the BoostAdapt algorithm, this section presents a simplified version of Boost and some discussion on related algorithms.

This version operates in $O(\log_{\frac{1}{1-\epsilon}}(\frac{k^2}{4\epsilon}))$ iterations. It adapts greedy threshold Badanidiyuru & Vondrák (2014) to update two disjoint sets $X$ and $Y$. In particular, it alternately selects elements with the marginal gain at least $\tau_X$ ($\tau_Y$) into $X$ ($Y$). Note that the thresholds $\tau_X$ and $\tau_Y$ are updated alternately after each iteration. Finally, the algorithm returns the best solution among $X$ and $Y$.

It is noted that our algorithm operates in a different way from Iterated Greedy Gupta et al. (2010), which works as follows: (1) it repeats greedy methods to construct two solutions: finding a feasible solution $X$ over ground set $V$ then finding another solution $Y$ over new ground set $V \setminus X$; (2) it then adapts the unconstrained submodular maximization (USM) algorithm to find the solution $X'$ over ground set $X$ and returns the best one among $X, Y$, and $X'$. Our algorithm also differs from the twin greedy strategy Han et al. (2020) and Interlaced Greedy Kuhnle (2019), allowing simultaneously updating two disjoint sets simultaneously in each iteration. In contrast to these algorithms, our algorithm allows the integration of ThreshSeq in each iteration; thus, it can be effectively parallelized.

---

**Algorithm 6:** Simplified Version of BoostAdapt Algorithm

---

1: **Input:** $f, V, k, \epsilon$
2: $M \leftarrow \max_{e \in V} f(e)$
3: $\Delta \leftarrow \lceil \log_{\frac{1}{1-\epsilon}}(\frac{k^2}{4\epsilon}) \rceil + 1$, $\tau \leftarrow kM/4$
4: $X \leftarrow \emptyset, Y \leftarrow \emptyset$
5: **for** $i \leftarrow 1$ to $\Delta$ **do**
6:   **if** $i$ is **odd then**
7:     $\tau_X \leftarrow \tau(1-\epsilon)^i$
8:     **for** $e \in V \setminus (X \cup Y)$ **do**
9:       **if** $f(e|X) \geq \tau_X$ **then**
10:         $X \leftarrow X \cup \{e\}$
11:   **else**
12:     $\tau_Y \leftarrow \tau(1-\epsilon)^i$
13:     **for** $e \in V \setminus (X \cup Y)$ **do**
14:       **if** $f(e|Y) \geq \tau_Y$ **then**
15:         $Y \leftarrow Y \cup \{e\}$
16: $S \leftarrow \arg\max_{Z \in \{X,Y\}} f(Z)$
17: **return** $S$

---

We define the following notations used for analyzing the algorithm's performance guarantees

- Supposing $X$ and $Y$ ordered: $X = \{x_1, x_2, \ldots, x_{|X|}\}, Y = \{y_1, y_2, \ldots, y_{|Y|}\}$, we conduct: $X^i = \{x_1, x_2, \ldots, x_i\}, Y^i = \{y_1, y_2, \ldots, y_i\}$, where $x_i$ (or $y_i$) is $i$-th element added into $X$ (or $Y$).

- For $e \in X \cup Y$, we assume that $x$ is added into $X$ or $Y$ at iteration $l(e)$.

- $X_i$ and $Y_i$ are $X$ and $Y$ after the iteration $i$ of the main loop and $X_0 = Y_0 = \emptyset$.

- For an element $e \in X \cup Y$, we denote $X^{<e}, Y^{<e}, \tau_X^e$ and $\tau_Y^e$ are $X, Y, \tau_X$ and $\tau_Y$ right before $e$ is selected into $X$ or $Y$, respectively.

- $\tau_X^{last}$ and $\tau_Y^{last}$ are $\tau_X$ and $\tau_Y$ at the last iterations when $X$ and $Y$ are considered to update.

We provide the performance guarantees of Algorithm 6 in Theorem B.1.

**Theorem B.1.** *For any $\epsilon \in (0, 1/4)$, the Algorithm 6 takes $O(\frac{n}{\epsilon} \log(\frac{k}{\epsilon}))$ query complexity and return an approximation ratio of $1/4 - \epsilon$.*

*Proof.* The Algorithm 6 takes $n$ queries to find $M$ and then runs in $\Delta = O(\log_{1/(1-\epsilon)}(\frac{k^2}{4\epsilon})) = O(\frac{1}{\epsilon} \log(\frac{k}{\epsilon}))$ iterations. Each iteration takes $O(n)$ queries to update $X$ or $Y$. Thus, the algorithm takes total $O(n) + O(\frac{n}{\epsilon} \log(\frac{k}{\epsilon})) = O(\frac{n}{\epsilon} \log(\frac{k}{\epsilon}))$.

For the approximation ratio. If $|X| \neq \emptyset$ or $|Y| \neq \emptyset$, i.e., it contains at least one element, we have

$$f(S) \geq \frac{(1-\epsilon)^2}{4} \max\{f(X_1), f(Y_2)\} \geq \frac{(1-\epsilon)^2 kM}{4} \geq \frac{(1-\epsilon)^2 \text{opt}}{4} > (\frac{1}{4} - \epsilon)\text{opt}. \quad (6)$$

We consider the other case when $X_1 = Y_2 = \emptyset$. In this case $f(X_1) = 0$, so $f(X_1 \cap O) = 0$. We further consider the following sub-cases

- If $|X| < k$ and $|Y| < k$. Considering an element $e$ is added into $Y$ (at iteration $l(e) \geq 4$), it was not added into $X$ at previous iterations. Therefore, it has marginal gain $f(e|X^{<e}) < \theta_X^e = \frac{\theta_Y^{l(e)}}{1-\epsilon} \leq \frac{f(e|Y^{<e})}{1-\epsilon}$. Combining this with $\frac{kM}{4} \geq \tau \geq \frac{(1-\epsilon)\epsilon M}{k}$, we have

$$f(O \cup X) - f(X) \leq \sum_{e \in O \setminus X} f(e|X) \quad (7)$$

$$= \sum_{e \in O \cap Y} f(e|X) + \sum_{e \in O \setminus (X \cup Y)} f(e|X) \quad (8)$$

$$\leq \sum_{e \in O \cap Y} \frac{f(e|Y^{<e})}{1-\epsilon} + \tau_X^{last} k \quad (9)$$

$$< \frac{f(Y)}{1-\epsilon} + \epsilon \text{opt} \quad (10)$$

where inequality equation 7 is due to the submodularity of $f$, inequality equation 9 is due to: $\tau_X^{last} \leq \epsilon M/k \leq \epsilon \text{opt}/k$, $f(Y) = \sum_{e \in Y} f(e|Y^{<e})$ and $Y$ contains elements with positive marginal gain.

On the other hand, since $X_1 = \emptyset$ any element $e$ is added into $X$ (at iteration $l(e) \geq 2$) having marginal gain $f(e|Y^{<e}) < \theta_Y^e = \theta_X^{l(e)-1} = \frac{\theta_Y^{l(e)}}{1-\epsilon} \leq \frac{f(e|X^{<e})}{1-\epsilon}$. Thus

$$f(O \cup Y) - f(Y) \leq \sum_{e \in O \cap X} f(e|Y) + \sum_{e \in O \setminus (X \cup Y)} f(e|X) \quad (11)$$

$$\leq \sum_{e \in O \cap X} \frac{f(e|X^{<e})}{1-\epsilon} + \tau_Y^{last} k \quad (12)$$

$$< \frac{f(X)}{1-\epsilon} + \epsilon \text{opt}. \quad (13)$$

By combining the submodularity of $f$, inequalities equation 10 and equation 13 and the selection rule of the final solution, we have:

$$f(O) \leq f(O \cup X) + f(O \cup Y) \quad (14)$$

$$\leq \frac{f(X) + f(Y)}{1-\epsilon} + f(X) + f(Y) + 2\epsilon \text{opt} \quad (15)$$

$$\leq 2\frac{2-\epsilon}{1-\epsilon} f(S) + 2\epsilon \text{opt} \quad (16)$$

which implies that

$$f(S) \geq \frac{(1-2\epsilon)(1-\epsilon)}{2(2-\epsilon)}\text{opt} > (\frac{1}{2}-\epsilon)(\frac{1}{2}-\frac{\epsilon}{2}) > (\frac{1}{4}-\epsilon)\text{opt}. \tag{17}$$

- If $|X| = k$ and $|Y| = k$, then $l(x_k) \geq 3$. Any element $e$ is not added into $X$ at iteration $l(x_k)$ is not added into $X$ at iteration $l(x_k) - 2$. On the other hand, since $|X| = k \geq |O|$, so $|X \setminus O| \geq |O \setminus X|$, we have:

$$f(X \cup O) - f(X) \leq \sum_{e \in O \setminus X} f(e|X) \tag{18}$$

$$\leq \sum_{e \in O \cap Y} f(e|X) + \sum_{e \in O \setminus (X \cup Y)} f(e|X) \tag{19}$$

$$< \frac{\sum_{e \in O \cap Y} f(e|Y^{<e})}{1-\epsilon} + |O \setminus X| \cdot \tau_X^{l(x_k)-2} \tag{20}$$

$$\leq \frac{\sum_{e \in O \cap Y} f(e|Y^{<e})}{1-\epsilon} + |X \setminus O| \cdot \frac{\tau_X^{l(x_k)}}{(1-\epsilon)^2} \tag{21}$$

$$\leq \frac{\sum_{e \in O \cap Y} f(e|Y^{<e})}{1-\epsilon} + \sum_{e \in X \setminus O} \frac{f(e|X^{<e})}{(1-\epsilon)^2}. \tag{22}$$

Similarly, we have

$$f(O \cup Y) - f(Y) \leq \sum_{e \in O \cap X} f(e|Y) + \sum_{e \in O \setminus (Y \cup Y)} f(e|X) \tag{23}$$

$$< \frac{\sum_{e \in O \cap X} f(e|X^{<e})}{1-\epsilon} + \tau_Y^{l(y_k)-2}k \tag{24}$$

$$= \frac{\sum_{e \in O \cap X} f(e|X^{<e})}{1-\epsilon} + \frac{\tau_Y^{l(y_k)}}{(1-\epsilon)^2}k \tag{25}$$

$$\leq \frac{\sum_{e \in O \cap X} f(e|X^{<e})}{1-\epsilon} + \sum_{e \in Y \setminus O} \frac{f(e|Y^{<e})}{(1-\epsilon)^2}. \tag{26}$$

By combining equation 22 and equation 26, we have

$$f(O) < \frac{\sum_{e \in O \cap Y} f(e|Y^{<e})}{1-\epsilon} + \sum_{e \in Y \setminus O} \frac{f(e|Y^{<e})}{(1-\epsilon)^2}$$

$$+ \sum_{e \in X \setminus O} \frac{f(e|X^{<e})}{(1-\epsilon)^2} + \frac{\sum_{e \in O \cap X} f(e|X^{<e})}{1-\epsilon} + f(X) + f(Y)$$

$$< \frac{f(X) + f(Y)}{(1-\epsilon)^2} + 2f(S) = (2 + \frac{2}{(1-\epsilon)^2})f(S) < \frac{4}{(1-\epsilon)^2}f(S)$$

which implies that $f(S) \geq \frac{(1-\epsilon)^2}{4}\text{opt} > (\frac{1}{4}-\epsilon)\text{opt}$.

- If $|X| < k$ and $|Y| = k$, by applying the transformations to equation 9 and equation 26 we have

$$f(O \cup X) - f(X) \leq \sum_{e \in O \cap Y} \frac{f(e|Y^{<e})}{1-\epsilon} + \epsilon\text{opt} \tag{27}$$

and

$$f(O \cup Y) - f(Y) \leq \frac{\sum_{e \in O \cap X} f(e|X^{<e})}{1-\epsilon} + \sum_{e \in Y \setminus O} \frac{f(e|Y^{<e})}{(1-\epsilon)^2}. \tag{28}$$

Therefore

$$f(O) \le f(O \cup X) + f(O \cup Y) \tag{29}$$

$$< \sum_{e \in O \cap Y} \frac{f(e|Y^{<e})}{1 - \epsilon} + \sum_{e \in Y \setminus O} \frac{f(e|Y^{<e})}{(1 - \epsilon)^2} \tag{30}$$

$$+ \frac{\sum_{e \in O \cap X} f(e|X^{<e})}{1 - \epsilon} + f(X) + f(Y) + \epsilon \mathsf{opt} \tag{31}$$

$$\le \frac{f(Y)}{(1 - \epsilon)^2} + \frac{f(X)}{1 - \epsilon} + 2f(S) + \epsilon \mathsf{opt} \tag{32}$$

$$< \frac{4}{(1 - \epsilon)^2} f(S) + \epsilon \mathsf{opt}. \tag{33}$$

Thus $f(S) \ge \frac{(1-\epsilon)^3}{4} \mathsf{opt} > (\frac{1}{4} - \epsilon)\mathsf{opt}$.

- If $|X| = k$ and $|Y| < k$, by applying the transformations to equation 12 and equation 22 we also have

$$f(O \cup Y) - f(Y) \le \sum_{e \in O \cap X} \frac{f(e|X^{<e})}{1 - \epsilon} + \epsilon \mathsf{opt} \tag{34}$$

and

$$f(O \cup X) - f(X) \le \frac{\sum_{e \in O \cap Y} f(e|Y^{<e})}{1 - \epsilon} + \sum_{e \in X \setminus O} \frac{f(e|X^{<e})}{(1 - \epsilon)^2}. \tag{35}$$

Therefore

$$f(O) \le f(O \cup X) + f(O \cup Y) \tag{36}$$

$$< \sum_{e \in O \cap X} \frac{f(e|X^{<e})}{1 - \epsilon} + \sum_{e \in X \setminus O} \frac{f(e|X^{<e})}{(1 - \epsilon)^2} \tag{37}$$

$$+ \frac{\sum_{e \in O \cap Y} f(e|Y^{<e})}{1 - \epsilon} + f(X) + f(Y) + \epsilon \mathsf{opt} \tag{38}$$

$$\le \frac{f(X)}{(1 - \epsilon)^2} + \frac{f(Y)}{1 - \epsilon} + 2f(S) + \epsilon \mathsf{opt} \tag{39}$$

$$< \frac{4}{(1 - \epsilon)^2} f(S) + \epsilon \mathsf{opt}. \tag{40}$$

Thus $f(S) \ge \frac{(1-\epsilon)^3}{4} \mathsf{opt} > (\frac{1}{4} - \epsilon)\mathsf{opt}$.

Combining all cases, we complete the proof. □

## C  PROBABILITY LEMMAS AND CONCENTRATION BOUNDS

This section provides Probability Lemmas and Concentration Bounds that are useful for analyzing the theoretical bounds of our algorithms.

**Lemma C.1** (Chernoff bounds Mitzenmacher & Upfal (2005)). *Supposing that $X_1, \ldots, X_n$ are independent binary random variables such that $\Pr[X_i = 1] = p_i$. Let $\mu = \sum_{i=1}^{n} p_i$, and $X = \sum_{i=1}^{n} X_i$. Then for any $\delta \ge 0$, we have*

$$\Pr[X \ge (1 + \delta)\mu] \le e^{\frac{-\delta^2 \mu}{2 + \delta}}. \tag{41}$$

*For $0 \le \delta \le 1$, we have*

$$\Pr[X \le (1 - \delta)\mu] \le e^{\frac{-\delta^2 \mu}{2}}. \tag{42}$$

**Lemma C.2** (Lemma 6 in Chen et al. (2021)). *Suppose there is a sequence of $n$ Bernoulli trials: $X_1, X_2, \ldots, X_n$, where the success probability of $X_i$ depends on the preceding trials $X_1, X_2, \ldots, X_{i-1}$. Suppose it holds that*

$$\Pr[X_i = 1 | X_1 = x_1, X_2 = x_2, \ldots, X_{i-1} = x_{i-1}] \geq \eta, \tag{43}$$

*where $\eta > 0$ is a constant and $x_1, \ldots, x_{i-1}$ are arbitrary.*
*Then if $Y_1, \ldots, Y_n$ are independent Bernoulli trials, each with probability $\eta$ of success, then*

$$\Pr\left(\sum_{i=1}^n X_i \leq b\right) \leq \Pr\left(\sum_{i=1}^n Y_i \leq b\right), \tag{44}$$

*where $b$ is an arbitrary integer. Moreover, let $A$ be the first occurrence of success in sequence $X_i$. Then,*

$$\mathbb{E}[A] \leq 1/\eta. \tag{45}$$

**Lemma C.3** (Lemma 7 in Chen et al. (2021)). *Suppose there is a sequence of $n + 1$ Bernoulli trials: $X_1, X_2, \ldots, X_{n+1}$, where the success probability of $X_i$ depends on the preceding trials $X_1, X_2, \ldots, X_{i-1}$, and it decreases from 1 to 0. Let $t = \max\{i : \Pr[X_i = 1] \geq \eta\}$, where $0 < \eta < 1$ is a constant. Then, it holds that*

$$\Pr[X_i = 1 | X_1 = x_1, X_2 = x_2, \ldots, X_{i-1} = x_{i-1}, i \leq t] \geq \eta, \tag{46}$$

*where $x_1, \ldots, x_{i-1}$ are arbitrary.*
*Then if $Y_1, \ldots, Y_{n+1}$ are independent Bernoulli trials, each with probability $\eta$ of success, then*

$$\Pr\left(\sum_{i=1}^t X_i \leq bt\right) \leq \Pr\left(\sum_{i=1}^t Y_i \leq bt\right), \tag{47}$$

*where $b$ is an arbitrary integer.*

**Lemma C.4** (Wald's Equation Wald (1945)). *Let $(X_n)_{n \in \mathbb{N}}$ be an infinite sequence of real-valued random variables and let $N$ be a nonnegative integer-valued random variable. Assume that: 1) $(X_n)_{n \in \mathbb{N}}$ are integrable (finite-mean) random variables, 2) $\mathbb{E}[X_n 1_{N \geq n}] = \mathbb{E}[X_n]P(N \geq n)$ for every natural number $n$, 3) the infinite series satisfies $\sum_{n=1}^\infty \mathbb{E}[|X_n|1_{\{N \geq n\}}] < \infty$. Then the random sums $S_N = \sum_{n=1}^N X_n$ and $T_N = \sum_{n=1}^N \mathbb{E}[X_n]$ are integrable and $\mathbb{E}[S_N] = \mathbb{E}[T_N]$.*

**Lemma C.5** (Lemma 11 in Chen et al. (2021)). *Let $(Y_i)$ be a sequence of independent and identically distributed Bernoulli trials, where the success probability is $\beta\epsilon$. Then for a constant integer $\alpha$, we have*

$$\Pr[\sum_{i=1}^\alpha Y_i > \epsilon\alpha] \leq \min\{\beta, e^{-\frac{(1-\beta)^2}{1+\beta}\epsilon\alpha}\}. \tag{48}$$

# D ALGORITHMS MISSED IN SECTION 3

In this section, we recap some subroutines in our algorithms and their performance bounds.

## D.1 ThreshSeq (ALGORITHM 2 IN CHEN & KUHNLE (2024))

## D.2 UNCONSTRAINED SUBMODULAR MAXIMIZATION ALGORITHMS

In the LinAdapt algorithm, we adapt the Fahrbach et al. (2019a)'s algorithm (USM1) that provides a factor of $1/4 - \epsilon$ with constant probability for the USM problem. The guarantees for the USM1 algorithm are sated in Lemma D.1.

**Lemma D.1.** *For any nonnegative submodular function $f$ and subset $A \subseteq V$, Algorithm 8 outputs a set $S \subseteq V$ in one adaptive round using $O(\log(1/\delta)/\varepsilon)$ oracle queries such that with probability at least $1 - \delta$ we have $f(S) \geq (1/4 - \varepsilon)\mathsf{opt}_A$, where $\mathsf{opt}_A = \max_{T \subseteq A} f(T)$.*

We further adapt an essentially optimal algorithm of Chen et al. (2019) for USM in our LBA which allows us to improve the approximation ratio slightly.

**Theorem D.2** (Chen et al. (2019)). *There is an algorithm that achieves a $1/(2 + \epsilon)$-approximation for USM using $O(\log(1/\epsilon)/\epsilon)$ adaptive rounds and $O(n \log^3(1/\epsilon)/\epsilon^4)$ query complexity.*

**Algorithm 7:** ThreshSeq$(V, \epsilon, \delta)$

1: **Input:** $f, V, k, \epsilon, \delta, \tau$
2: $A \leftarrow \emptyset, A' \leftarrow \emptyset, U \leftarrow V, \Delta = \lceil 4(\frac{2}{\epsilon} \log(n) + \log(\frac{n}{\delta})) \rceil$
3: **for** $j \leftarrow 1$ to $\Delta$ **do**
4:     Update $U \leftarrow \{x \in U : f(x|A) \geq \tau\}$
5:     **if** $|U| = 0$ **then**
6:         **return** $A, A'$
7:     $U \leftarrow \mathbf{random - permutation}(U)$
8:     $s \leftarrow \min\{k - |A|, |V|\}$
9:     $B[1:s] \leftarrow [none, none, \ldots, none]$
10:     **for** $i \leftarrow 1$ to $s$ (**in parallel**) **do**
11:        $T_{i-1} \leftarrow \{v_1, v_2, \ldots, v_{i-1}\}$
12:        **if** $f(v_i|A \cup T_{i-1}) \geq \tau$ **then**
13:           $B[i] \leftarrow \mathbf{true}$
14:        **else**
15:           **if** $f(v_i|A \cup T_{i-1}) < 0$ **then**
16:              $B[i] \leftarrow \mathbf{false}$
17:     $i^* \leftarrow \max\{i : \#\text{trues in } B[1:i] \geq (1-\epsilon)i\}$
18:     $A \leftarrow A \cup T_{i^*}$
19:     $A' \leftarrow A' \cup T_{i^*}[\text{where B} \neq \mathbf{false}]$
20:     **if** $|A| = k$ **then**
21:         **return** $A, A'$
22: **return** *failure*

**Algorithm 8:** USM1$(V, \epsilon, \delta)$

1: **Input:** $f$, a subset $A \subseteq V, \varepsilon, \delta$
2: Set iteration bound $t \leftarrow \lceil \log(1/\delta) / \log(1 + (4/3)\varepsilon) \rceil$
3: **for** $i = 1$ to $t$ in parallel **do**
4:     Let $R_i$ be a uniformly random subset of $A$
5: Set $S \leftarrow \arg\max_{X \in \{R_1, \ldots, R_t\}} f(X)$
6: **return** $S$

# E    PROOFS MISSED OF SECTION 4

## E.1    PROOFS MISSED OF LinBoundSet (ALGORITHM 2)

*Proof of Lemma 4.2.* If $S \leq k$, $S' = S$, thus the Lemma holds. We consider the remaining case $S > k$. Let $\overline{S} = S \setminus S'$. Beside the notations of $T_i, T'_{\lambda_i}$ in Line 9 of Algorithm 2, we use the following useful notations:

- $S_j$ and $T_{j,\lambda_j^*}$ are $S$ and $T_{\lambda^*}$ after iteration $j$.

- $T_{j,i}$ is the set of first $i$ elements added into $T_{j,\lambda_j^*}$.

- $T'_{j,\lambda_i} = T_{j,\lambda_i} \setminus T_{j,\lambda_{i-1}}$.

- $T''_{\lambda_i} = \{v_j \in T'_{\lambda_i} : b[j] \neq \textbf{false}\}$, i.e., elements in $T'_{\lambda_i}$ with non-negative marginal gain.

- $T^j = \bigcup_{\lambda_i \leq \lambda_j^*} T''_{\lambda_i}$; $T^j_{\lambda_i} = \bigcup_{\lambda \leq \lambda_i} T''_\lambda$.

- $c = \max\{c \in \mathbb{N} : S' \subseteq \bigcup_{j=c}^{\ell} T^j\}$.

By the selection rule in Line 20, Algorithm 2, $T''_{\lambda_i}$ only contains elements with non-negative marginal gain. If $T'_{\lambda_i}$ is a **good** block, it has at least $(1 - \epsilon)|T'_{\lambda_i}|$ good elements and thus $|T''_{\lambda_i}| \geq (1 - \epsilon)|T'_{\lambda_i}|$. It's easy to see that $|T^j| < k$ for any $j > c$, and the size of $T^c$ may be larger than $k$. Therefore, we consider two following cases:

**Case 1.** If $j > c$, $|T^j| \leq k$. By the selection rule of $\lambda^*$ in Line 19, Algorithm 2, there are $\lambda_j^* - 1$ first blocks in $T_{j,\lambda_j^*}$ are good and the last one is bad. For any block $T'_{j,\lambda_i} \subseteq T_{j,\lambda_j^*}, \lambda_i < \lambda_j^*$, we have

$$
\begin{aligned}
f(T''_{\lambda_i}|S_{j-1} \cup T^j_{\lambda_{i-1}}) &\geq f(T''_{\lambda_i}|S_{j-1} \cup T_{j,\lambda_{i-1}}) &&\text{(By the submodularity and } T^j_{\lambda_{i-1}} \subseteq T_{j,\lambda_{i-1}}) \\
&\geq \sum_{v_l \in T''_{\lambda_i}} f(v_l|S_{j-1} \cup T_{j,l-1}) \\
&\geq (1 - \epsilon)\alpha|T'_{j,\lambda_i}|\frac{M_{\lambda_{i-1}}}{k} &&\text{(Since } T'_{j,\lambda_i} \text{ is a good block)} \\
&\geq (1 - \epsilon)\alpha|T'_{j,\lambda_i}|\frac{f(S_{j-1})}{k} &&\text{(By selection rule of } M_{\lambda_{i-1}}).
\end{aligned}
\tag{49}
$$

Since the block size exponentially increases with factor of $1 + \epsilon$ when $\lambda_i \leq k$, so we have $|T'_{j,\lambda_j^*}| \leq \epsilon|T_{j,\lambda_j^*}|/(1 + \epsilon)$. Combining this with equation 49, we have:

$$
\begin{aligned}
f(T^j|S_{j-1}) &= \sum_{\lambda_i \leq \lambda_j^*} f(T''_{\lambda_i}|S_{j-1} \cup T^j_{\lambda_{i-1}}) &&(50) \\
&\geq \sum_{\lambda_i < \lambda_j^*} f(T''_{\lambda_i}|S_{j-1} \cup T^j_{\lambda_{i-1}}) \quad \text{(Since } b[e] \geq 0, \forall e \in T''_{\lambda_i}) &&(51) \\
&\geq \sum_{\lambda_i < \lambda_j^*} (1 - \epsilon)\alpha|T'_{j,\lambda_i}|\frac{f(S_{j-1})}{k} &&(52) \\
&= (1 - \epsilon)\alpha|T_{j,\lambda_j^*} \setminus T'_{j,\lambda_j^*}|\frac{f(S_{j-1})}{k} &&(53) \\
&\geq \frac{1 - \epsilon}{1 + \epsilon}\alpha\frac{|T_{j,\lambda_j^*}|}{k}f(S_{j-1}) &&(54) \\
&\geq \frac{1 - \epsilon}{1 + \epsilon}\alpha\frac{|T^j|}{k}f(S_{j-1}). &&(55)
\end{aligned}
$$

**Case 2.** If $j = c$, also by the selection rule of $\lambda^*$, the last block $T'_{c,\lambda_c^*}$ is always bad. If $|T_{c,\lambda_c^*}| \leq k$, all previous blocks $T'_{c,\lambda_i}, \lambda_i < \lambda_c^*$ are good. If $|T_{c,\lambda_c^*}| > k$, some previous contiguous blocks are

good and they contain at least $k$ elements. By the selection rule of $S$, we can let $\lambda_r = \min\{\lambda_r \in \Lambda : \bigcup_{\lambda_r \leq \lambda_i \leq \lambda_c^*} T''_{c,\lambda_i} \subseteq S'\}$. With a note that each block $T''_{c,\lambda_i}$ contains at most $\epsilon k$ elements and $S'$ is the union of some blocks, we have:

$$f(T^c|\overline{S}) = \sum_{\lambda_r \leq \lambda_i \leq \lambda_c^*} f(T''_{\lambda_i}|\overline{S} \cup T^c_{\lambda_{i-1}}) \tag{56}$$

$$\geq \sum_{\lambda_r \leq \lambda_i < \lambda_c^*} f(T''_{\lambda_i}|\overline{S} \cup T_{c,\lambda_{i-1}}) \tag{57}$$

$$\geq \sum_{\lambda_r \leq \lambda_i < \lambda_c^*} (1-\epsilon)\alpha|T'_{c,\lambda_i}|\frac{M_{\lambda_{i-1}}}{k} \tag{58}$$

$$\geq \sum_{\lambda_r \leq \lambda_i < \lambda_c^*} (1-\epsilon)\alpha|T'_{c,\lambda_i}|\frac{f(\overline{S})}{k} \tag{59}$$

$$\geq (1-\epsilon)\alpha|(T_{c,\lambda_j^*} \cap S') \setminus T'_{c,\lambda_j^*}|\frac{f(\overline{S})}{k} \tag{60}$$

$$\geq (1-\epsilon)\alpha \cdot \max\{0, |T^c \cap S'| - \epsilon k\}\frac{f(\overline{S})}{k}. \tag{61}$$

Combining two cases above with a note that $k \geq |S'| \geq (1-\epsilon)k$, we have:

$$f(S) - f(\overline{S}) = f(T^c|\overline{S}) + \sum_{j=c+1}^{\ell} f(T^j|S_{j-1}) \tag{62}$$

$$\geq (1-\epsilon)\alpha \cdot \max\{0, |T^c \cap S'| - \epsilon k\}\frac{f(\overline{S})}{k} + \sum_{j=c+1}^{\ell} \frac{1-\epsilon}{1+\epsilon}\alpha\frac{|T^j|}{k}f(S_{j-1}). \tag{63}$$

$$\geq (1-\epsilon)\alpha \cdot \max\{0, |T^c \cap S'| - \epsilon k\}\frac{f(\overline{S})}{k} + \frac{1-\epsilon}{1+\epsilon}\alpha\frac{|S' \setminus T^c|}{k}f(\overline{S}). \tag{64}$$

$$\geq \frac{1-\epsilon}{1+\epsilon}\alpha(|S'| - \epsilon k)\frac{f(\overline{S})}{k} \tag{65}$$

$$\geq \frac{1-\epsilon}{1+\epsilon}\alpha(k - 2\epsilon k)\frac{f(\overline{S})}{k} \tag{66}$$

$$\geq \frac{1-\epsilon}{1+\epsilon}(1-2\epsilon)\alpha f(\overline{S}) \tag{67}$$

which implies that $f(S) \geq (1 + \frac{(1-\epsilon)(1-2\epsilon)\alpha}{1+\epsilon})f(\overline{S})$. By the submodularity of $f$, we have

$$f(S) \leq f(S') + f(\overline{S}) \implies f(S') \geq f(S) - f(\overline{S}) \geq \frac{(1-\epsilon)(1-2\epsilon)\alpha}{1+\epsilon+(1-\epsilon)(1-2\epsilon)\alpha}f(S) \tag{68}$$

which implies the proof. □

*Proof of Lemma 4.3.* We first show that the following propositions are true.

**Proposition E.1.** *For each iteration in* LinBoundSet, *let* $\lambda_{(t)} = \max\{\lambda_i \in \Lambda : \lambda_i < t\}$ *and* $Q_t = \{e \in V : f(e|S \cup T_t) < \alpha M_{\lambda_{(t)}}/k\}$, *we have* $|Q_0| = 0$, $Q_{|V|} = |V|$, *and* $|Q_t| \leq |Q_{t+1}|$.

By Line 4 of Algorithm 2, we have $f(e|S) \geq \alpha f(S)/k$ for all $e \in V$. Therefore $|Q_0| = 0$. On the other hand, since $f(e|S \cup V) = 0 < \alpha M_{\lambda_{(|V|)}}/k$ for all $e \in V$, $|Q_{|V|}| = |V|$. For the last property, by the submodularity of $f$ and the definition of $M_{\lambda_i}$, for any element $e \in Q_t$ we have

$$f(e|S \cup T_{t+1}) \leq f(e|S \cup T_t) < \alpha\frac{M_{\lambda_{(t)}}}{k} \leq \alpha\frac{M_{\lambda_{(t+1)}}}{k}, \tag{69}$$

which implies that $e \in Q_{t+1}$. Hence $Q_t \subseteq Q_{t+1}$ and $|Q_t| \leq |Q_{t+1}|$.

Now, we provide a bound probability that $B[\lambda]$ is false in the following Proposition.

**Proposition E.2.** *Let $t = \min\{j \in \mathbb{N} : |Q_j| \geq \beta\epsilon|V|\}$, $\lambda_{(t)} = \max\{\lambda \in \Lambda : \lambda < t\}$ and $(Y_j)$ be the sequence of the independent and identically distributed Bernoulli trials with the success probability is $\beta\epsilon$. For any $\lambda < \lambda_{(t)}$, we have $\Pr[B[\lambda] = \mathbf{false}] \leq \Pr[\sum_{j=1}^{|T'_\lambda|} Y_j > \epsilon|T'_\lambda|]$.*

We define an element $v_i$ that is **bad** if $f(v_i|S \cup T_{i-1}) < \alpha M_{\lambda_{(i)}}/k$ and **good** otherwise. Consider the random permutation of $V$ as a sequence of independent Bernoulli trials, with success if the element is bad and failure otherwise.

We have $\Pr[v_i \text{ is } \mathbf{bad}|v_1, \ldots, v_{i-1}] = |Q_{i-1}|/|V|$ thus the probability that $v_j, j < t$ is bad is less than $\beta\epsilon$. By the definition of $B[\lambda]$ in Line 16, Algorithm 2, the block $T'_\lambda$ is bad, it must contain more than $\epsilon|T'_\lambda|$ elements. Let $X_i = 1$ if $v_i$ is bad; and $X_i = 0$, otherwise, we have

$$\Pr(B[\lambda] = \mathbf{false}) \leq \Pr(\text{the number of bad elements in } T'_\lambda > \epsilon|T'_\lambda|) \tag{70}$$

$$= \Pr\left(\sum_{v_j \in T'_\lambda} X_j > \epsilon|T'_\lambda|\right) \tag{71}$$

$$\leq \Pr\left(\sum_{j=1}^{|T'_\lambda|} Y_j > \epsilon|T'_\lambda|\right) \quad \text{(Due to appplying Lemma C.3).} \tag{72}$$

The probability iteration $j$ failure is upper bounded by the probability $\lambda^* < t$. We now consider two cases. If $\lambda^* \leq k$, there is at least one block $T'_\lambda$ with $\lambda \in \Lambda, \lambda < \lambda_{(t)}$ is bad. If $\lambda^* > k$, let $\lambda'_{(t)} = \max\{\lambda' \in \Lambda : \sum_{\lambda \in \Lambda, \lambda' \leq \lambda \leq \lambda_{(t)}} |T'_\lambda| \geq k\}$. Then there must be at least one integer $\lambda \in \Lambda$ between $\lambda_{(t)}$ and $\lambda'_{(t)}$ such that block $T'_\lambda$ is bad. Therefore, let $B_1 = \{\lambda \in \Lambda : \lambda < k\}$, $B_2 = \{\lambda \in \Lambda : |\Lambda \cap [\lambda, \lambda_{(t)}]| \leq \lceil 1/\epsilon \rceil\}$, then we have

$$\Pr(\text{iteration } j \text{ fails}) \leq \Pr[\lambda^* < t] \leq \Pr(\exists \lambda \in B_1 \cup B_2, B[\lambda] = \mathbf{false}) \leq \frac{1}{2}. \tag{73}$$

The proof of inequality equation 73 is based on some properties of Bernoulli trials, which is shown in detail in Chen et al. (2021). For the sake of completeness, we write down the details of the proof below.
By Lemma C.4 we have

$$\Pr[\text{iteration } j \text{ fails}] \leq \Pr[\exists \lambda \in B_1 \cup B_2 \text{ with } B[\lambda] = \mathbf{false}] \tag{74}$$

$$= \mathbb{E}[1_{\{\exists \lambda \in B_1 \cup B_2 \text{ with } B[\lambda]=\mathbf{false}\}}] \tag{75}$$

$$\leq \mathbb{E}\left[\sum_{\lambda \in B_1} 1_{\{B[\lambda]=\mathbf{false}\}} + \sum_{\lambda \in B_2} 1_{\{B[\lambda]=\mathbf{false}\}}\right] \tag{76}$$

$$\leq \mathbb{E}\left[\sum_{\lambda \in B_1} 1_{\{B[\lambda]=\mathbf{false}\}}\right] + \mathbb{E}\left[\sum_{\lambda \in B_2} 1_{\{B[\lambda]=\mathbf{false}\}}\right] \tag{77}$$

$$= \mathbb{E}\left[\sum_{\lambda \in B_1} \mathbb{E}[1_{\{B[\lambda]=\mathbf{false}]\}}]\right] + \mathbb{E}\left[\sum_{\lambda \in B_2} \mathbb{E}[1_{\{B[\lambda]=\mathbf{false}\}}]\right] \tag{78}$$

where equation 78 holds, since the sequence $(1_{\{B[\lambda]=\mathbf{false}\}})$ and the random variable $t$ follows the assumptions in Lemma C.4: 1) $1_{\{B[\lambda]=\mathbf{false}\}}$s are all integrable random variables because they only take the values 0 and 1; 2) $t$ is a stopping time since it only depends on the previous $t-1$ selections; 3) $\Pr[1_{\{t \geq n\}} = 0)] = 1$ for any $n \geq |V|$.

The first term of *equation* 78 is bounded by adapting Proposition E.2 as follows:

$$\mathbb{E}\left[\sum_{\lambda \in B_1} \mathbb{E}[1_{\{B[\lambda]=\mathbf{false}\}}]\right] \leq \mathbb{E}\left[\sum_{\lambda \in B_1} \leq \Pr\left(\sum_{j=1}^{|T'_\lambda|} Y_j > \epsilon|T'_\lambda|\right)\right] \tag{79}$$

$$\leq \mathbb{E}\left[\sum_{\lambda \in \{\lfloor(1+\epsilon)^u\rfloor:u\geq 1\}} \Pr\left(\sum_{j=1}^{|T'_\lambda|} Y_j > \epsilon|T'_\lambda|\right)\right] \tag{80}$$

$$\leq \sum_{\lambda \in \{\lfloor(1+\epsilon)^u\rfloor:u\geq 1\}} \min\{\beta, e^{-\frac{(1-\epsilon)^2}{1+\beta}\epsilon|T'_\lambda|}\} \text{ (By Lemma C.5)} \tag{81}$$

$$\leq \sum_{\lambda \geq 1} \min\{\beta, e^{-\frac{\epsilon\lambda}{2}}\} \text{ (Due to } \beta < 1/2, |T'_\lambda| < \lambda) \tag{82}$$

$$\leq \alpha\beta + \sum_{\lambda=a+1}^{\infty} e^{-\frac{\epsilon\lambda}{2}}, \text{ (where } a = \lfloor\frac{1}{8\beta}\rfloor = \lfloor\frac{2}{\epsilon}\log(\frac{8}{1-e^{-\epsilon/2}})\rfloor) \tag{83}$$

$$\leq \alpha\beta + \frac{e^{-\epsilon(a+1)/2}}{1 - e^{-\epsilon/2}} \tag{84}$$

$$\leq \frac{1}{8} + \frac{1}{8} = \frac{1}{4}, \tag{85}$$

where equation 80 follows from $B_1 \subseteq \{\lfloor(1+\epsilon)^u : u \geq 1\rfloor\}$, and let $\{\lfloor(1+\epsilon)^u : u \geq 1\rfloor\}\} = \{\lambda_1, \lambda_2, \ldots\}, |T'| = \lambda_i - \lambda_{i-1}$;

For the second term in equation 78, from Proposition E.2, Lemma C.5 and the fact that $|B_2| < \lceil 1/\epsilon\rceil$, we have

$$\mathbb{E}\left[\sum_{\lambda \in B_2} \mathbb{E}[1_{\{B[\lambda]=\mathbf{false}\}}]\right] \leq \mathbb{E}\left[\sum_{\lambda \in B_2} \leq \Pr\left(\sum_{j=1}^{|T'_\lambda|} Y_j > \epsilon|T'_\lambda|\right)\right] \leq \beta\lceil 1/\epsilon\rceil \leq \frac{1}{4} \tag{86}$$

Put it into equation 78, we have

$$\Pr[\text{iteration } j \text{ fails}] < \frac{1}{4} + \frac{1}{4} = \frac{1}{2} \tag{87}$$

$\square$

*Proof of Theorem 4.4.* **Prove the success probability.** Since each successful iteration will remove $(\beta\epsilon)$-fraction of $V$ thus if the LinBoundSet fails, there are no more than $\lceil\log_{1/(1-\beta\epsilon)}(n)\rceil$ successful iterations. Let $X$ be the number of successes in the $\ell$ iterations. Then, By Lemma 4.3, $X$ can be regarded as a sum of dependent Bernoulli trials, where the success probability is larger than $1/2$. Let $Y$ be a sum of independent Bernoulli trials, where the success probability is equal to $1/2$. By applying Lemma C.5, we have

$$\Pr[\text{ LinBoundSet \textbf{fails}}] \leq \Pr[X \leq \lceil\log_{1/(1-\beta\epsilon)}(n)\rceil] \tag{88}$$

$$\leq \Pr[Y \leq \lceil\log_{1/(1-\beta\epsilon)}(n)\rceil] \quad \text{(By Lemma C.5)} \tag{89}$$

$$\leq \Pr\left[Y \leq \frac{\log(n)}{\beta\epsilon}\right] \tag{90}$$

$$\leq e^{-\frac{1}{2}(\frac{(2\beta\epsilon+1)^2}{2(\beta\epsilon+1)})^2 2(1+\frac{1}{\beta\epsilon})\log(\frac{n}{\delta})} \quad \text{(By Lemma C.1)} \tag{91}$$

$$\leq (\frac{\delta}{n})^{\frac{(2\beta\epsilon+1)^2}{4\beta\epsilon(\beta\epsilon+1)}} \tag{92}$$

$$\leq \frac{\delta}{n}. \tag{93}$$

**Prove the bound of** $f(X \cup S)$. We use the notation in the proof of Lemma 4.2. We now further prove that $f(X \cup S) \leq (\alpha + 1)f(S)$. For each element $x \in X \setminus S$, define $j(x) + 1$ be the iteration

where $x$ is filtered out (Line 4, Algorithm 2), we have $f(x|S_{j(x)}) \leq \alpha f(S_{j(x)})/k$. Therefore.

$$f(X \cup S) - f(S) \leq \sum_{x \in X \setminus S} f(x|S_{j(x)}) \tag{94}$$

$$\leq \sum_{x \in X \setminus S} \alpha \frac{f(S_{j(x)})}{k} \tag{95}$$

$$\leq \alpha f(S_{j(x)}) \qquad \text{(Sice } |X| \leq k) \tag{96}$$

$$\leq \alpha f(S). \tag{97}$$

Combine this with equation 68, we obtain

$$f(X \cup S) \leq (\alpha + 1)f(S) \leq (\alpha + 1)\left(1 + \frac{1 + \epsilon}{(1 - \epsilon)(1 - 2\epsilon)\alpha}\right)f(S') \tag{98}$$

$$= \left(2 + \alpha + \frac{1}{\alpha} + (1 + \frac{1}{\alpha})\frac{2(2 - \epsilon)}{(1 - \epsilon)(1 - 2\epsilon)}\epsilon\right)f(S') \tag{99}$$

**Prove the adaptive and query complexities.** LinBoundSet requires the oracle queries on Lines 4, 10, 12 of Algorithm 2. At these times, the queries are executed in parallel, there are constant adaptive rounds in each iteration or the main loop. Thus, the algorithm needs $O(\ell) = O(\log(n/\delta)/\epsilon^3)$ adaptive complexity.

Let $V_j$ be the set $V$ after Line 4 at iteration $j$ and let $Y_i$ be the number of iterations between $(i-1)$-th success and $i$-th success. By Lemma C.2, we have $\mathbb{E}[Y_i] \leq 2$. At each iteration $j$, the algorithm takes $|V_{j-1}| + 1$ queries on Line 4, takes $|\Lambda| \leq |V_j|$ queries on Line 10 and takes total $|V_j|$ queries after the second and the third loops. Therefore, the number of queries is bounded by

$$\sum_{j=1}^{\ell} \mathbb{E}(|V_{j-1}| + 1 + 2|V_j|) \leq n + \ell + \sum_{j=1}^{\ell} 3|V_j| \tag{100}$$

$$\leq n + \ell + 3\sum_{i=1}^{\infty}[Y_i(1 - \epsilon\beta)^i n] \tag{101}$$

$$\leq n + \ell + 3\frac{n}{\beta\epsilon} \tag{102}$$

$$= O(\frac{n}{\epsilon^3}). \tag{103}$$

This completes the proof. $\qquad\square$

# F   PROOFS MISSED OF SECTION 5

## F.1   PROOFS MISSED OF LinAst (ALGORITHM 3)

LinAst calls LinAdapt to find $S_0$. This task takes $O(\frac{\log n}{\epsilon^2})$ adaptivity and $O(\frac{n}{\epsilon^3})$ queries. Then, the algorithm runs in $2\ell = 2(\lceil \log_{1-\epsilon}(\frac{1}{a}) \rceil + 1) = O(\frac{1}{\epsilon}\log(\frac{1}{\epsilon}))$ iterations in one adaptive round. Each iteration takes $O(\frac{n}{\epsilon})$ query complexity and $O(\frac{1}{\epsilon}\log(\frac{n}{\delta}))$ adaptive rounds to call ThreshSeq algorithm. Therefore, the adaptive complexity of LinAst is at most

$$O(\frac{\log n}{\epsilon^2}) + O(\frac{1}{\epsilon}\log(\frac{n}{\delta})) = O(\frac{\log(n)}{\epsilon^2}) \tag{104}$$

and its query complexity is

$$O(\frac{n}{\epsilon^3}) + O(\frac{1}{\epsilon}\log(\frac{1}{\epsilon})\frac{n}{\epsilon}) = O(\frac{n}{\epsilon^3}). \tag{105}$$

The probability that LinAdapt fails is at most $1/(3n)$. In the main loop of LinAst, each iteration returns sets $(A_i, A_i')$ (or $(A_i, A_i')$) with fail probability at most $1/(3n)$. Therefore, the algorithm returns the set $C$ as the final solution with a successful probability of at least $1 - 1/n$ by the union

bound. By Theorem 4.1, we have $f(S_0) \leq \text{opt} \leq af(S_0)$, where $a = (12 + (\frac{16}{1-4\epsilon} + \frac{8(2-\epsilon)}{(1-\epsilon)(1-2\epsilon)})\epsilon)$. Thus $\frac{f(S_0)}{ck} \leq \frac{\text{opt}}{ck} \leq \frac{af(S_0)}{ck} = M$. Besides, $M(1-\epsilon)^i \in [\frac{f(S_0)}{ck}, \frac{af(S_0)}{ck}]$. Therefore, there exist an integer $i$ so that

$$\frac{(1-\epsilon)\text{opt}}{ck} \leq M(1-\epsilon)^i \leq \frac{\text{opt}}{ck}. \tag{106}$$

If the algorithm succeeds, the proof of approximation ratio is similar to the proof of Theorem 7 in Chen & Kuhnle (2024). For the sake of completeness, we write down the details of the proof by following. Denote $\tau_i$ is the value of $\tau$ at iteration $i$ in LinAst.

**Case 1:** If $|A| = k$ or $|B| = k$. By the theoretical guarantee of ThreshSeq (Theorem 3.2), we have

$$f(C) \geq \max\{f(A'), f(B')\} \geq (1-\epsilon)\tau_i \tag{107}$$

$$\geq \frac{(1-\epsilon)^2}{c}\text{opt} \geq (\frac{1}{6} - \epsilon)\text{opt}. \tag{108}$$

**Case 2:** If $|A| < k$ and $|B| < k$. By Theorem 3.2, for any element $e \in V$ we have $f(e|A) \leq \tau_i$ and $f(e|B) \leq \tau_i$. By the submodularity of $f$, we have

$$f(O \cup A) - f(A) \leq \sum_{e \in O} f(e|A) < k\tau_i \leq \frac{\text{opt}}{c}. \tag{109}$$

$$f((O \setminus A) \cup B) - f(B) \leq \sum_{e \in O \setminus A} f(e|B) < k\tau_i \leq \frac{\text{opt}}{c}. \tag{110}$$

Combine two inequalities equation 109-equation 110 with the fact that $A \cap B = \emptyset$, we have

$$f(A') + f(B') \geq f(A) + f(B) \tag{111}$$

$$\geq f(O \cup A) + f((O \setminus A) \cup B) - \frac{2\text{opt}}{c} \tag{112}$$

$$\geq f(O \setminus A) + f(O \cup A \cup B) - \frac{2\text{opt}}{c} \quad \text{(Due to the submodularity)} \tag{113}$$

$$\geq f(O \setminus A) - \frac{2\text{opt}}{c} \quad \text{(Due to the non-negativity)} \tag{114}$$

Besides, the USM2 gives approximation ratio of $1/(2+\epsilon)$, we have $\mathbb{E}[f(A'')] \geq f(O \cup A)$. Combine this with equation 114, we have

$$\text{opt} \leq f(O) \leq f(O \cup A) + f(O \setminus A) \tag{115}$$

$$\leq (2+\epsilon)\mathbb{E}[f(C)] + 2f(C) + \frac{2\text{opt}}{c}. \tag{116}$$

it follows that $\mathbb{E}[f(C)] \geq \frac{\text{opt}}{c} \geq (\frac{1}{6} - \epsilon)\text{opt}$. The proof is completed.

### F.2 Proofs missed of LinAtg (Algorithm 4)

*Proof of Theorem 5.1.* **Prove the complexities.** The algorithm LinAdapt takes $O(\frac{\log n}{\epsilon^2})$ adaptivity and $O(\frac{n}{\epsilon^3})$ queries to call LinAdapt. Then, the algorithm runs two loops, each loop takes $\ell = \log_{\frac{1}{1-\epsilon'}}(ac)] + 1 = O(\frac{1}{\epsilon}\log(\frac{1}{\epsilon}))$ sequential iterations. In each iteration, it takes $O(\frac{n}{\epsilon})$ query complexity and $O(\frac{1}{\epsilon}\log(\frac{n}{\delta}))$ to call ThreshSeq algorithm, where $\delta = 1/(3\ell)$. Therefore, the adaptive complexity LBA is

$$O(\frac{\log n}{\epsilon^2}) + O(\frac{1}{\epsilon}\log(\frac{1}{\epsilon})\frac{1}{\epsilon}\log(\frac{n}{\delta})) = O\Big(\frac{1}{\epsilon^2}\log(\frac{1}{\epsilon})\log(\frac{n}{\epsilon}\log(\frac{1}{\epsilon}))\Big) \tag{117}$$

$$= O\Big(\frac{1}{\epsilon^2}\log(\frac{1}{\epsilon})\big(\log(n) + \log(\frac{1}{\epsilon}\log(\frac{1}{\epsilon}))\big)\Big) \tag{118}$$

$$= O\Big(\frac{\log(n)}{\epsilon^2}\log(\frac{1}{\epsilon})\Big). \tag{119}$$

and its query complexity is

$$O(\frac{n}{\epsilon^3}) + O(\frac{1}{\epsilon}\log(\frac{1}{\epsilon})\frac{n}{\epsilon}) = O(\frac{n}{\epsilon^3}). \tag{120}$$

The probability that LinAdapt fails is at most $1/n$, and the probability that ThreshSeq fails in each iteration is at most $\delta$. By the union probability, the probability BoostAdapt fails is at most $1/n + 2\ell \cdot \delta = 1/n$.

**Prove the approximation ratio.** ATG algorithm in Chen & Kuhnle (2024) use $f(e_{max})$ to give the bound $f(O)/k \leq f(e_{max}) \leq f(O)$. Our LinAtg uses $M = af(S_0)/k$ to give a bound of $\text{opt}/k$. By Theorem 4.1, we have $f(S_0) \leq \text{opt} \leq af(S_0)$, where $a = (12 + (\frac{16}{1-4\epsilon} + \frac{8(2-\epsilon)}{(1-\epsilon)(1-2\epsilon)})\epsilon)$. Thus

$$\frac{M}{a} = \frac{f(S_0)}{k} \leq \frac{\text{opt}}{k} \leq \frac{af(S_0)}{k} = M \tag{121}$$

implying that $M(1-\epsilon')^i \in [\frac{M}{ca}, M] \supseteq [\frac{\text{opt}}{ck}, \frac{\text{opt}}{k}]$. From the main loops, LBA works similarly to ATG. Therefore, we follow the proof of Theorem 8 in Chen & Kuhnle (2024). For the sake of completeness, we write down the details of the proof below. The notations used in the proof are listed below.

- $A'$ and $A'$ are returned by the first loop of Algorithm 4, while $B$ and $B'$ are returned by the second one.

- Define $A_i$ as $A$ after iteration $i$.

- Let $\mathcal{A}'_j$ be the first $j$ elements in $A$, where $1 \leq j \leq |A|$. Furthermore, for $|A'| < j \leq k$, let $\mathcal{A}'_j$ be $A'$ combined with $k - |A'|$ dummy elements.

- Let $\{a'_j\} = \mathcal{A}'_j \setminus \mathcal{A}'_{j-1}$, $a'_j$ be returned at iteration $i(j)$, and $A_{i(j)}$ be the set $A$ returned at iteration $i(j)$. If $\{a'_j\}$ is dummy element, let $i(j) = \ell + 1$. Then, we likewise define $B_i$, $\mathcal{B}'_j$ $B_{i(j)}$.

**Lemma F.1** (Lemma 9 in Chen & Kuhnle (2024)). *For $1 \leq j \leq k$, there are at least $\lceil(1-\epsilon')k\rceil$ of $j$ such that*

$$f(\mathcal{A}'_j) - f(\mathcal{A}'_{j-1}) + \frac{M}{ck} \geq \frac{1-\epsilon}{k}(f(O \cup A_{i(j)-1}) - f(\mathcal{A}'_{j-1})). \tag{122}$$

*And for any $j$,*

$$f(\mathcal{A}'_j) \geq f(\mathcal{A}'_{j-1}). \tag{123}$$

**Lemma F.2** (Lemma 10 in Chen & Kuhnle (2024)). *For $1 \leq j \leq k$, there are at least $\lceil(1-\epsilon')k\rceil$ of $j$ such that*

$$f(\mathcal{B}'_j) - f(\mathcal{B}'_{j-1}) + \frac{M}{ck} \geq \frac{1-\epsilon}{k}(f((O \setminus A) \cup B_{i(j)-1}) - f(\mathcal{B}'_{j-1})). \tag{124}$$

*And for any $j$,*

$$f(\mathcal{B}'_j) \geq f(\mathcal{B}'_{j-1}). \tag{125}$$

**Lemma F.3** (Lemma 11 in Chen & Kuhnle (2024)). *Let $\Gamma_u = f(\mathcal{A}'_{j(u)}) + f(\mathcal{B}'_{j(u)})$, where $j(u)$ is the $u$-th $j$ which satisfies Lemma F.1 or Lemma F.2. Then, there are at least $\lceil(1-\epsilon')k\rceil$ of $u$ follow that*

$$f(O \setminus A) - \Gamma_u - \frac{2M}{c(1-\epsilon')} \leq \left(1 - \frac{1-\epsilon'}{k}\right)\left(f(O \setminus A) - \Gamma_{u-1} - \frac{2M}{c(1-\epsilon')}\right). \tag{126}$$

Lemma F.3 yields a recurrence of the form $(b - u_{i+1}) \leq a(b - u_i)$, $u_0 = 0$, and has a solution $u_i \geq b(1 - a^i)$. Consequently, we have

$$f(A') + f(B') \geq \left(1 - \left(1 - \frac{1-\epsilon'}{k}\right)^{(1-\epsilon')k}\right)\left(f(O \setminus A) - \frac{2M}{c(1-\epsilon')}\right) \tag{127}$$

$$\geq \left(1 - e^{-(1-\epsilon')^2}\right)\left(f(O \setminus A) - \frac{2M}{c(1-\epsilon')}\right). \tag{128}$$

Let $\beta = 1 - e^{-(1-\epsilon')^2}$. From the choice of $C$ on line 16, Algorithm 4, we have $2f(C) \geq f(A') + f(B')$ and from equation 128, we have

$$f(O \setminus A) \leq \frac{2}{\beta}f(C) + \frac{2M}{c(1-\epsilon')^2} \tag{129}$$

$$\leq \frac{2}{\beta}f(C) + \frac{2f(O)}{c(1-\epsilon')^2}. \tag{130}$$

Assume USM2 has a ratio of $1/\rho$ for USM problem. For any $A$, $f(O \cap A)/\rho \leq \mathbb{E}[f(A'')|A]$; therefore

$$f(O \cap A) \leq \rho\mathbb{E}[f(C)]. \tag{131}$$

For any set $A$, $f(O) \leq f(O \cap A) + f(O \setminus A)$ by the submodularity and nonnegativity. Thus

$$f(O) \leq f(O \cap A) + f(O \setminus A) \tag{132}$$

$$\leq \frac{2}{\beta}f(C) + \frac{2f(O)}{c(1-\epsilon')^2} + \rho. \tag{133}$$

Therefore

$$f(C) \geq \frac{1 - \frac{2}{c(1-\epsilon')}}{\rho + \frac{2}{\beta}}f(O) \geq \left(\frac{e-1}{\rho(e-1)+2e} - \epsilon\right)f(O). \tag{134}$$

By replacing $\rho = 2 + \epsilon$ for USM2, we obtain the ratio. $\qquad \square$

## G   PROOFS MISSED OF SECTION 6

We recap the following notations used for analyzing the BoostAdapt's performance guarantees

- Supposing $X$ and $Y$ ordered: $X = \{x_1, x_2, \ldots, x_{|X|}\}, Y = \{y_1, y_2, \ldots, y_{|Y|}\}$, we conduct: $X^i = \{x_1, x_2, \ldots, x_i\}, Y^i = \{y_1, y_2, \ldots, y_i\}$, where $x_i$ (or $y_i$) is $i$-th element added into $X$ (or $Y$).

- For $e \in X \cup Y$, we assume that $e$ is added into $X$ or $Y$ at iteration $i_e$.

- $X_i$ and $Y_i$ are $X$ and $Y$ after the iteration $i$ of the main loop and $X_0 = Y_0 = \emptyset$. Then, we define $X_i'$ and $Y_i'$ analogously.

- $\tau_X^i$ and $\tau_Y^i$ are $\tau_X$ and $\tau_Y$ after the iteration $i$ of the main loop of BoostAdapt.

- $\tau_X^{last}$ and $\tau_Y^{last}$ are $\tau_X$ and $\tau_Y$ at the last iterations of the main loop of BoostAdapt when $X$ and $Y$ are considered to update.

- Define $A_i^+$ (res. $B_i^+$) as the set of elements in $A_i$ (res. $B_i$) having the marginal gain at least $\tau_X$ (res. $\tau_Y$) at iteration $i$, $X_i^+ = \cup_{j=1}^i A_i^+, Y_i^+ = \cup_{j=1}^i B_i^+$ and $X^+ = X_\Delta^+, Y^+ = Y_\Delta^+$.

- For an element $e \in X \cup Y$, we denote by $X^{<e}, Y^{<e}$ as $X, Y$ right before $e$ is selected into $X$ or $Y$, respectively.

We first explore some basic properties of $X, X', X''$ and $Y, Y', Y''$ in Lemma G.1.

**Lemma G.1.** *a) For any iteration $i$, we have $f(X_i') \geq f(X_i)$ and $f(Y_i') \geq f(Y_i)$.*
*b) At the end of* BoostAdapt, *we have $f(X'') \geq (1 - \epsilon)f(X')$ and $f(Y'') \geq (1 - \epsilon)f(Y')$.*

*Proof of G.1.* **a)** We prove the Lemma by the induction hypothesis. For $i = 1$, we have $f(X_1') = f(A_1') \geq f(A_1) = f(X_1)$ due to the Theorem 3.2, so the Proposition holds. Assuming the proposition holds for $j$, we have:

$$f(X_{j+1}') = f(A_{j+1}' \cup X_j') = f(A_{j+1}'|X_j') + f(X_j') \tag{135}$$

$$\geq f(A_{j+1}'|X_j) + f(X_j) \tag{136}$$

$$\geq f(A_{j+1}|X_j) + f(X_j) = f(X_{j+1}) \tag{137}$$

where the inequality equation 136 is due to the submodularity of $f$ and $X'_j \subseteq X_j$, inequality equation 137 is due to the theoretical bound of ThreshSeq (Theorem 3.2). Therefore, by the induction hypothesis, we have $f(X'_i) \geq f(X_i)$ for any iteration $i$.

By similar reasoning, we can prove $f(Y'_i) \geq f(Y_i)$ for any iteration $i$, which completes the proof of **a)**.

**b)**. If $|X'| = k$, then $X'' = X'$ the Lemma holds. If $|X'| > k$, by the setting of $k'$ and the selection rule of $X''$, we get $|X''| = k \geq k'(1 - \epsilon) \geq |X|(1 - \epsilon) \geq |X'|(1 - \epsilon)$. Therefore $|X' \setminus X''| \leq \epsilon |X'|$. Elements in $|X'' \setminus X'|$ have lowest marginal so we obtain

$$f(X') = \sum_{e \in X'} f(e|X'^{<e}) = \sum_{e \in X''} f(e|X'^{<e}) + \sum_{e \in X' \setminus X''} f(e|X'^{<e}) \tag{138}$$

$$\leq \sum_{e \in X''} f(e|X'^{<e}) + \epsilon \sum_{e \in X'} f(e|X'^{<e}) \tag{139}$$

$$\leq \sum_{e \in X''} f(e|X''^{<e}) + \epsilon f(X') \quad \text{(Due to the submodularity and } X'' \subseteq X') \tag{140}$$

$$= f(X'') + \epsilon f(X') \tag{141}$$

which implies $f(X'') \geq (1 - \epsilon)f(X')$. By similar reasoning, we can prove $f(Y'') \geq (1 - \epsilon)f(Y')$. This completes the proof of **b)**. $\square$

*Proof of Lemma 6.1.* **a)** For $i \geq 1$, $i$ is odd. If $i = 1$, $X_1 = Y_0 = \emptyset$. Thus $C = \emptyset$ and the Lemma holds. If $i \geq 3$, let $C = C_2 \cup C_4 \cup \ldots \cup C_{i-1}$, where $C_j$ $(j \leq i - 1)$ is the set of elements in $C$ added into $Y_j^+$, i.e., $C_j = B_j^+ \cap C$. Each element $e \in C_j$ is selected into $Y_j^+$ at iteration $j \leq i - 1$ and has not been added yet into $X$ at the previous iteration. By the Theorem 3.2 and the submodularity of $f$, for any $e \in C_j$ we have

$$f(e|X_i) \leq f(e|X_{j-1}) < \tau_X^{j-1} = \frac{\tau_Y^j}{1 - \epsilon} \leq \frac{f(e|Y^{<e})}{1 - \epsilon}. \tag{142}$$

It follows that

$$\sum_{e \in C} f(e|X_i) = \sum_{j=2,4,\ldots,i-1} \sum_{e \in C_j} f(e|X_i) \tag{143}$$

$$< \sum_{j=2,4,\ldots,i-1} \sum_{e \in C_j} \frac{f(e|Y^{<e})}{1 - \epsilon} \tag{144}$$

$$= \sum_{e \in C} \frac{f(e|Y^{<e})}{1 - \epsilon}. \tag{145}$$

**b)** We prove this case by the same reasoning with part **a)**
If $i \geq 2$, $i$ is even. If $i = 2$, since $X_1 = \emptyset$, thus $D = \emptyset$ and the Lemma holds. If $i \geq 4$, let $D = D_1 \cup D_3 \cup \ldots \cup D_{i-1}$, where $D_j = A_j^+ \cap D$. Each $e \in D$ is selected into $X$ at iteration $j, 3 \leq j \leq i - 1$ and has not been added yet into $Y$ at the previous iteration. By the Theorem 3.2, for any $e \in D_j$ we have

$$f(e|Y_i) \leq f(e|Y_{j-1}) < \tau_Y^{j-1} = \frac{\tau_X^j}{1 - \epsilon} \leq \frac{f(e|X^{<e})}{1 - \epsilon} \tag{146}$$

Therefore

$$\sum_{e \in D} f(e|Y_i) = \sum_{j=3,5,\ldots,i-1} \sum_{e \in D_j} f(e|Y_i) \tag{147}$$

$$< \sum_{j=3,5,\ldots,i-1} \sum_{e \in D_j} \frac{f(e|X^{<e})}{1 - \epsilon} \tag{148}$$

$$\leq \sum_{e \in D} \frac{f(e|X^{<e})}{1 - \epsilon} \tag{149}$$

which completes the proof. $\square$

*Proof of Lemma 6.2.* We consider following cases:

**Case 1.** If $|X| < k'$ and $|Y| < k'$, by the definition of $\Delta$ in Algorithm 5, we have $\frac{M}{4k} \geq \tau(1-\epsilon)^i \geq \frac{\epsilon(1-\epsilon)f(S_0)}{k}$. Thus $\tau_X^{last} \leq \epsilon f(S_0)/k \leq \epsilon \mathsf{opt}/k$. By the submodularity and applying Lemma 6.1 and Theorem 3.2, we have

$$f(X \cup O) - f(X) \leq \sum_{e \in O \setminus X} f(e|X) \tag{150}$$

$$= \sum_{e \in Y^+ \cap O} f(e|X) + \sum_{e \in O \setminus (X \cup Y^+)} f(e|X) \tag{151}$$

$$< \frac{\sum_{e \in Y^+ \cap O} f(e|Y^{<e})}{1-\epsilon} + k\tau_X^{last} \tag{152}$$

$$\leq \frac{\sum_{e \in Y' \cap O} f(e|Y^{<e})}{1-\epsilon} + k\tau_X^{last} \tag{153}$$

$$\leq \frac{f(Y')}{1-\epsilon} + \epsilon \mathsf{opt} \tag{154}$$

where inequality equation 150 is due to the submodularity of $f$, inequality equation 152 is due to: applying Lemma 6.1 with $D = Y^+ \cap O$ and applying Theorem 3.2.

Similarity, since $X_1 = \emptyset$ and by applying Lemma 6.1 and Theorem 3.2 we have

$$f(Y \cup O) - f(Y) \leq \sum_{e \in O \setminus Y} f(e|Y) \quad \text{(Due to the submodularity of } f) \tag{155}$$

$$= \sum_{e \in O \cap X^+} f(e|Y) + \sum_{e \in O \setminus (X^+ \cup Y)} f(e|Y) \tag{156}$$

$$< \frac{\sum_{e \in O \cap X^+} f(e|X^{<e})}{1-\epsilon} + k\tau_Y^{last} \tag{157}$$

$$\leq \frac{\sum_{e \in O \cap X'} f(e|X^{<e})}{1-\epsilon} + k\tau_Y^{last} \tag{158}$$

$$\leq \frac{f(X')}{1-\epsilon} + \epsilon \mathsf{opt}. \tag{159}$$

Combining two inequalities equation 154 and equation 159 and the selection rule of the final solution we have

$$f(O) \leq f(O \cup X) + f(O \cup Y) \quad \text{(Due to the submodularity of } f) \tag{160}$$

$$< \frac{f(X') + f(Y')}{1-\epsilon} + f(X) + f(Y) + 2\epsilon \mathsf{opt} \tag{161}$$

$$= \frac{2-\epsilon}{1-\epsilon}\Big(f(X') + f(Y')\Big) + 2\epsilon \mathsf{opt} \tag{162}$$

$$\leq \frac{2-\epsilon}{(1-\epsilon)^2}\Big(f(X'') + f(Y'')\Big) + 2\epsilon \mathsf{opt} \quad \text{(Due to Lemma C.2)} \tag{163}$$

$$\leq \frac{4-2\epsilon}{(1-\epsilon)^2}f(S) + 2\epsilon \mathsf{opt} \tag{164}$$

**which implies the proof of a).**

**Case 2.** If $|X| = k'$ and $|Y| = k'$. Since $X_1 = 0$ we must have $i_{x_{k'}} > 1$. If $i_{y_{k'}} = 2$, we have

$$f(S) \geq f(Y_2') \geq (1-\epsilon)|Y_2|\tau_Y^2 \geq \frac{(1-\epsilon)^3 \mathsf{opt}}{4}. \tag{165}$$

thus the Lemma holds.

If $i_{y_{k'}} > 2$, then each element $e$ that is not added into $X$ at iteration $i_{x_{k'}}$ also has not been added into $X$ at iteration $i_{x_{k'}} - 2$. By applying Lemma 6.1 with a note that $|X_{i_{x_{k'}}-2}| < k'$, we have

$f(e|X) < \tau_X^{i_{x_{k'}}-2}$. Therefore

$$f(X \cup O) - f(X) \le \sum_{e \in O \setminus X} f(e|X) \tag{166}$$

$$\le \sum_{e \in Y^+ \cap O} f(e|X) + \sum_{e \in O \setminus (X \cup Y^+_{i_{x_{k'}}-1})} f(e|X^{<e}) \tag{167}$$

$$< \frac{\sum_{e \in Y^+ \cap O} f(e|Y^{<e})}{1 - \epsilon} + |O \setminus X| \cdot \tau_X^{i_{x_{k'}}-2}. \tag{168}$$

Similarly, each element $e$ is not added to $Y$ at iteration $i_{y_{k'}}$ also is not added to $Y$ at iteration $i_{y_{k'}} - 2$. By Lemma 6.1, we have:

$$f(Y \cup O) - f(Y) \le \sum_{e \in O \cap X^+} f(e|Y) + \sum_{e \in O \setminus (Y \cup X^+_{i_{y_{k'}}-1})} f(e|Y) \tag{169}$$

$$\le \frac{\sum_{e \in O \cap X^+} f(e|X^{<e})}{1 - \epsilon} + |O \setminus Y| \cdot \tau_Y^{i_{y_{k'}}-2}. \tag{170}$$

On the other hand, since $|X| = k' > k \ge |O|$. Since at each iteration $i$, ThreshSeq selects at least $(1 - \epsilon)|A_i|$ element with marginal gain is above the threshold, so $|A_i^+| \ge (1 - \epsilon)|A_i|$. Thus $|X^+| \ge (1 - \epsilon)|X| = (1 - \epsilon)k'$. If $(1 - \epsilon)k'$ is an integer, then $(1 - \epsilon)k' = k$ and thus $|X^+| = k$. If $(1 - \epsilon)k'$ is not an integer, then $(1 - \epsilon)k' < k$, $\lceil (1 - \epsilon)k' \rceil = k$. Since $|X^+|$ is an integer, we have $|X^+| = \lceil (1 - \epsilon)k' \rceil = k$. Overall, $|X^+| = k$ which implies $|X^+ \setminus O| \ge |O \setminus X^+| \ge |O \setminus X'| \ge |O \setminus X|$. By the definition of $X', X^+$ we have

$$\sum_{e \in X^+ \setminus O} f(e|X^{<e}) \ge |X^+ \setminus O| \cdot \tau_X^{i_{x_{k'}}} \tag{171}$$

$$\ge (1 - \epsilon)^2 |O \setminus X^+| \tau_X^{i_{x_{k'}}-2} \tag{172}$$

$$\ge (1 - \epsilon)^2 |O \setminus X| \tau_X^{i_{x_{k'}}-2} \tag{173}$$

$$\implies |O \setminus X| \tau_X^{i_{x_{k'}}-2} \le \frac{\sum_{e \in X^+ \setminus O} f(e|X^{<e})}{(1 - \epsilon)^2}. \tag{174}$$

Similarly, since $|Y| = k'$, $|Y^+| \ge k$ and $|Y^+ \setminus O| \ge |O \setminus Y^+|$ we have

$$\sum_{e \in Y^+ \setminus O} f(e|Y^{<e}) \ge (1 - \epsilon)^2 |O \setminus Y| \tau_Y^{i_{y_k}-2} \tag{175}$$

$$\implies |O \setminus Y| \tau_Y^{i_{y_k}-2} \le \frac{\sum_{e \in Y^+ \setminus O} f(e|Y^{<e})}{(1 - \epsilon)^2}. \tag{176}$$

Combining equation 168, equation 170, equation 174 and equation 176 together we get

$$f(O) \leq f(O \cup X) + f(O \cup Y) \tag{177}$$

$$\leq \frac{\sum_{e \in Y^+ \cap O} f(e|Y^{<e})}{1 - \epsilon} + \frac{\sum_{e \in Y^+ \setminus O} f(e|Y^{<e})}{(1 - \epsilon)^2} \tag{178}$$

$$+ \frac{\sum_{e \in O \cap X^+} f(e|X^{<e})}{1 - \epsilon} + \frac{\sum_{e \in X^+ \setminus O} f(e|X^{<e})}{(1 - \epsilon)^2} + f(X) + f(Y) \tag{179}$$

$$\leq \frac{\sum_{e \in Y^+} f(e|Y^{<e})}{(1 - \epsilon)^2} + \frac{\sum_{e \in X^+} f(e|X^{<e})}{(1 - \epsilon)^2} + f(X) + f(Y) \tag{180}$$

$$\leq \frac{\sum_{e \in Y'} f(e|Y^{<e})}{(1 - \epsilon)^2} + \frac{\sum_{e \in X'} f(e|X^{<e})}{(1 - \epsilon)^2} + f(X) + f(Y) \tag{181}$$

$$\leq \frac{f(Y')}{(1 - \epsilon)^2} + \frac{f(X')}{(1 - \epsilon)^2} + f(X) + f(Y) \tag{182}$$

$$< \frac{2f(Y')}{(1 - \epsilon)^2} + \frac{2f(X')}{(1 - \epsilon)^2} = 2\frac{f(X') + f(Y')}{(1 - \epsilon)^2} \tag{183}$$

$$\leq 2\frac{f(X'') + f(Y'')}{(1 - \epsilon)^3} \quad \text{(Due to Lemma C.2)} \tag{184}$$

**Case 3.** If $|Y| = k'$ and $|X| < k'$. If $i_{y_k} = 2$, we have

$$f(S) \geq f(Y_2) \geq (1 - \epsilon)^2 \tau_Y^2 \geq (1 - \epsilon)^2 \frac{\text{opt}}{4} \tag{185}$$

the Lemma holds. If $i_{y_k} > 2$, similar to the **Case 2** of this proof, we also have

$$f(Y \cup O) - f(Y) \leq \frac{\sum_{e \in O \cap X^+} f(e|X^{<e})}{1 - \epsilon} + |O \setminus Y| \cdot \tau_Y^{i_{y_{k'}} - 2}. \tag{186}$$

and

$$|O \setminus Y|\tau_Y^{i_{y_k} - 2} \leq \frac{\sum_{e \in Y^+ \setminus O} f(e|Y^{<e})}{(1 - \epsilon)^2}. \tag{187}$$

Since $|X| < k'$, similar to the **Case 1**, we have

$$f(X \cup O) - f(X) \leq \sum_{e \in O \setminus X} f(e|X) \tag{188}$$

$$= \sum_{e \in Y^+ \cap O} f(e|X) + \sum_{e \in O \setminus (X \cup Y^+)} f(e|X) \tag{189}$$

$$\leq \frac{\sum_{e \in Y^+ \cap O} f(e|Y^{<e})}{1 - \epsilon} + k\tau_X^{last}. \tag{190}$$

It follows that

$$f(O) \leq f(O \cup Y) + f(O \cup X) \tag{191}$$

$$< \frac{\sum_{e \in Y^+ \cap O} f(e|Y^{<e})}{1 - \epsilon} + \frac{\sum_{e \in Y^+ \setminus O} f(e|Y^{<e})}{(1 - \epsilon)^2} + \frac{\sum_{e \in O \cap X^+} f(e|X^{<e})}{1 - \epsilon} \tag{192}$$

$$+ f(X) + f(Y) + \epsilon\text{opt} \tag{193}$$

$$\leq \frac{\sum_{e \in Y^+} f(e|Y^{<e})}{(1 - \epsilon)^2} + \frac{\sum_{e \in X'} f(e|X^{<e})}{1 - \epsilon} + f(X) + f(Y) + \epsilon\text{opt} \tag{194}$$

$$< \frac{f(Y')}{(1 - \epsilon)^2} + \frac{f(X')}{1 - \epsilon} + f(X) + f(Y) + \epsilon\text{opt} \tag{195}$$

$$< 2\frac{f(Y') + f(Y')}{(1 - \epsilon)^2} + \epsilon\text{opt} \tag{196}$$

$$\leq 2\frac{f(X'') + f(Y'')}{(1 - \epsilon)^3} + \epsilon\text{opt} \tag{197}$$

**Case 4.** If $|Y| < k'$ and $|X| = k'$. Since $X_1 \neq 0$ we must have $i_{x_k} > 1$. Similar to the **Case 2**, we have

$$f(X \cup O) - f(X) \leq \sum_{e \in O \setminus X} f(e|X) < \frac{\sum_{e \in Y^+ \cap O} f(e|Y^{<e})}{1 - \epsilon} + |O \setminus X| \cdot \tau_X^{i_{x_{k'}} - 2} \qquad (198)$$

and

$$|O \setminus X| \tau_X^{i_{x_{k'}} - 2} \leq \frac{\sum_{e \in X^+ \setminus O} f(e|X^{<e})}{(1 - \epsilon)^2}. \qquad (199)$$

Since $|Y| < k'$ similar the **Case 1**, we have

$$f(Y \cup O) - f(Y) \leq \frac{\sum_{e \in O \cap X^+} f(e|X^{<e})}{1 - \epsilon} + |O \setminus Y| \cdot \tau_Y^{i_{y_{k'}} - 2} \qquad (200)$$

It follows that

$$f(O) \leq f(O \cup X) + f(O \cup Y) \qquad (201)$$

$$\leq \frac{\sum_{e \in Y^+ \cap O} f(e|Y^{<e})}{1 - \epsilon} + \frac{\sum_{e \in O \cap X^+} f(e|X^{<e})}{1 - \epsilon} \qquad (202)$$

$$+ \frac{\sum_{e \in X^+ \setminus O} f(e|X^{<e})}{(1 - \epsilon)^2} + f(X) + f(Y) + \epsilon \text{opt} \qquad (203)$$

$$\leq \frac{\sum_{e \in Y'} f(e|Y^{<e})}{1 - \epsilon} + \frac{\sum_{e \in X^+} f(e|X^{<e})}{(1 - \epsilon)^2} \qquad (204)$$

$$\leq \frac{f(Y')}{1 - \epsilon} + \frac{f(X')}{(1 - \epsilon)^2} + f(X) + f(Y) + \epsilon \text{opt}. \qquad (205)$$

$$\leq 2 \frac{f(X') + f(Y')}{(1 - \epsilon)^2} + \epsilon \text{opt}. \qquad (206)$$

$$\leq 2 \frac{f(X'') + f(Y'')}{(1 - \epsilon)^3} + \epsilon \text{opt}. \qquad (207)$$

Combine **Case 2**, **Case 3**, **Case 4**, we have:

$$f(O) \leq 2 \frac{f(X'') + f(Y'')}{(1 - \epsilon)^4} \leq \frac{4f(S)}{(1 - \epsilon)^4} \qquad (208)$$

**which completes the proof of b).** $\qquad \square$

*Proof of Theorem 6.3.* **Prove successful probability and complexities.** BoostAdapt first calls LinAdapt to find $S_0$ in $O(\frac{\log n}{\epsilon^2})$ adaptivity and $O(\frac{n}{\epsilon^3})$ queries. Then, the algorithm runs in $\Delta = O(\frac{1}{\epsilon} \log(\frac{k}{\epsilon}))$ iterations; each takes $O(\frac{n}{\epsilon})$ query complexity and $O(\frac{1}{\epsilon} \log(\frac{n}{\delta}))$ adaptive rounds to call ThreshSeq algorithm. Therefore, the adaptive complexity BoostAdapt is

$$O(\frac{\log n}{\epsilon^2}) + O\Big(\frac{1}{\epsilon} \log(\frac{k}{\epsilon}) \frac{1}{\epsilon} \log(\frac{n}{\delta})\Big) = O\Big(\frac{1}{\epsilon^2} \log(\frac{k}{\epsilon}) \log(\frac{n}{\epsilon} \log(\frac{k}{\epsilon}))\Big) \qquad (209)$$

$$= O\Big(\frac{1}{\epsilon^2} \log(\frac{k}{\epsilon}) \log(\frac{n}{\epsilon}) + \frac{1}{\epsilon^2} \log(\frac{k}{\epsilon}) \log(\frac{k}{\epsilon}))\Big) \qquad (210)$$

$$= O\Big(\frac{1}{\epsilon^2} \log(\frac{k}{\epsilon}) \log(\frac{n}{\epsilon})\Big) \qquad (211)$$

and its query complexity is

$$O(\frac{n}{\epsilon^3}) + O(\frac{1}{\epsilon} \log(\frac{k}{\epsilon}) \frac{n}{\epsilon}) = O(\frac{n}{\epsilon^2} \log(\frac{k}{\epsilon})) \qquad (212)$$

The probability that LinAdapt fails is at most $1/(3n)$, and the probability that ThreshSeq fails in each iteration is at most $\delta$. By the union probability, the probability BoostAdapt fails is at most $1/(3n) + \Delta \cdot \delta = 1/n$. $\qquad \square$

# H ADDITIONAL EXPERIMENTAL DETAILS AND RESULTS

In this section, we elaborate on the experimental configuration and provide additional empirical findings along with some discussions.

## H.1 ALGORITHMS AND SETTINGS

We compare our algorithms LinAst, LinAtg and BoostAdapt with state-of-the-arts for non-monotone SMC including:

- **IteratedGreedy (IG)**: The algorithm in Gupta et al. (2010) achieves $1/6 - \epsilon$ approximation ratio when the $1/2$-approximation of Buchbinder et al. (2012) is used for the unconstrained maximization subproblem. The algorithm takes $O(nk)$ query and adaptive complexities.

- **AdaptiveNonmonotoneMax (ANM)**: The algorithm in Fahrbach et al. (2023) achieves $0.039 - \epsilon$ approximation ratio with $O(\log(n))$ adaptivity complexity and $O(n \log(k))$ query complexity.

- ParSKP2: The algorithm in Cui et al. (2023) that runs in $O(\log^2 n)$ adaptivity, $O(nk \log^2 n)$ query complexity and returns a $1/4 - \epsilon$ approximation solution in expectation.

- **AdaptiveSimpleThreshold (AST)**: The algorithm in Chen & Kuhnle (2024) achieves $1/6 - \epsilon$ approximation ratio with $O(\log(n/\delta)/\epsilon + \log(1/\epsilon)/\epsilon)$ adaptivity complexity and $O(\log_{1-\epsilon}(1/(6k)).(n/\epsilon + n \log^3(1/\epsilon)/\epsilon^4))$ query complexity.

- **AdaptiveThresholdGreedy (ATG)**: The algorithm in Chen & Kuhnle (2024) achieves $0.193 - \epsilon$ approximation ratio with $O(\log(n) \log(k))$ adaptivity complexity and $O(n \log(k))$ query complexity.

- ENE: The algorithm of Ene and Nguyen Ene & Nguyen (2020) that returns a ratio of $1/e - \epsilon$ in $O(\log n)$ adaptive rounds and $\Omega(nk^2 \log n)$.

- **FastRandomGreedy (FRG)**: The algorithm in Buchbinder et al. (2015) achieves $1/e - \epsilon$ approximation ratio with $O(k)$ adaptivity complexity and $O(n)$ query complexity.

The comparison is about **four metrics**: object values, adaptive complexity, number of queries, and running time. We experimented with two well-known applications: Revenue Maximization (RM) and Maximum Cut (MC) Chen & Kuhnle (2024); Kuhnle (2019); Amanatidis et al. (2020).

We set $\epsilon = 0.1$ for all algorithms, and other settings are set the same as Chen & Kuhnle (2024). Furthermore, we use an algorithm in Fahrbach et al. (2019a) (USM1), which returns a ratio of $1/4 - \epsilon$ for our LinAst, LinAtg and algorithms (AST, ATG, ANM) in Chen & Kuhnle (2024); Fahrbach et al. (2023) and set the same setting with Chen & Kuhnle (2024); Fahrbach et al. (2023). ENE algorithm requires access to an oracle for the multilinear extension and its gradient. In the case of maximum cut, the multilinear extension and its gradient can be computed in closed form in time linear in the size of the graph and thus one can evaluate it using direct oracle access to the multilinear extension and its gradient on the maximum cut application. However, no closed form exists for the multilinear extension of the revenue maximization objective Chen & Kuhnle (2024).

## H.2 APPLICATIONS AND SETTINGS

The applications utilized in the experiments are defined as follows:

**Maximum Cut Application.** Given graph $G = (V, E)$, and nonnegative edge weight $w_{ij}$ for each edge $(i, j) \in E$. For $S \subseteq V$, let

$$f(S) = \sum_{i \in V \setminus S} \sum_{j \in S} w_{ij}. \tag{213}$$

The objective of the problem is to find a solution set $S$ such that $f(S)$ is maximized while ensuring that the cardinality of $S$ is less than $k$. The function $f(\cdot)$ is a submodular and non-monotone function.

**Revenue Maximization Application.** Considering a graph $G = (V, E)$ representing a social network, where each edge $(i, j) \in E$ is associated with a non-negative weight $w_{ij}$, we adopt the concave graph model introduced by (Hartline et al., 2008). In this model, each user $i \in V$ is linked to a non-negative, a concave function $f_i : \mathbb{R}^+ \to \mathbb{R}^+$. The function $v_i(S) = f_i\left(\sum_{j \in S} w_{ij}\right)$ indicates the likelihood of user $i$ purchasing a product if the set $S$ adopts it. Thus, the total revenue from seeding a set $S$ is given by

$$f(S) = \sum_{i \in V \setminus S} f_i\left(\sum_{j \in S} w_{ij}\right). \tag{214}$$

The objective of the application is to find a solution $S$ such that $f(S)$ is maximized while ensuring that the cardinality of $S$ is less than $k$. The function $f(\cdot)$ is a submodular and non-monotone function.

Table 2: Details of datasets used in the experiments.

| Dataset Name | Nodes | Edges | Types | Applications | Sources\References |
|---|---|---|---|---|---|
| Barabási-Albert | 968 | 5,708 | Undirected | MC | Chen & Kuhnle (2024) |
| ca-GrQc | 5,242 | 14,496 | Undirected | MC | SNAP |
| ca-Astro | 18,772 | 198,110 | Undirected | RM | SNAP |
| web-Google | 875,713 | 5,105,039 | Undirected | MC | SNAP |

**Other settings.** In our experimental setup, we employed OpenMP to parallelize the code written in C++. To precisely measure the execution time of these algorithms, we utilized **Chrono**, a standard component of C++. Our methodology includes marking the start and end points of each algorithm's execution using Chrono and computing the time gap between these two points. In the Maximum Cut, we uniformly set the weight of all edges to $1.0$. In the Revenue Maximization, we randomly assigned weights to all edges from the interval $(0, 1)$. Besides, we experimented on a high-performance computing (HPC) server cluster with the following parameters: partition=large, #threads(CPU)=64, node=2, max memory = 3,073 GB.

### H.3 ADDITIONAL RESULTS

Figure 2 shows the results of compared algorithms on Barabási-Albert and Astro with both RM and MC, while Figure 3 focuses on the largest dataset, Google, with various $k$ on the MC application. On Google, we do not show the result of ENE and ParSKP2 because they become impractical with large size data and large $k$.

**Objective value:** Figures 2-3(a)(e) show the objective values. They show that our BoostAdapt lines consistently reach the highest points with every $k$. LinAtg approximates to BoostAdapt while LinAst is a little lower than BoostAdapt. With Barabási-Albert and Astro, except AST and ParSKP2, other algorithms are also consistent with ours. AST are moderately lower than ours. Significantly, ParSKP2 always hits the lowest values. With Google, the gap between ours with FRG, AST, and ParSKP2 seems larger, especially in the case of small $k$. Especially, Figure 3(a) indicates ours are much higher than FRG, AST and ParSKP2. In this case, our algorithms can be up to 1.3-1.6 times larger than FRG and AST. Finally, our algorithms significantly improve the solution quality.

**Adaptive rounds:** In RM (Figure 2(b)), ANM marks the lowest points, which means it saves the best adaptive rounds. Being consistent to ANM they are LinAst, FRG and AST. Moderately higher than these lines they are LinAtg, IG and BoostAdapt. Significantly, ParSKP, ENE and ATG are much higher than the others. In MC (Figures 2-3(f)), ATG, IG, and ParSKP2 always waste a sharply large number of adaptive rounds, the others can use an acceptable number of them. Especially, our LinAst always hits the lowest points while our other algorithms are almost close to it. Also, AST and FRG save more considerable rounds compared to mentioned high adaptivity lines. As can be seen, our algorithms outperform the others in the adaptive rounds.

**Number of queries:** In both RM and MC of Figures 2(c)(g), our algorithms almost always minimize the number of queries. In RM Figure 2(c), LinAtg, ENE, FRG, ParSKP2 and BoostAdapt are the

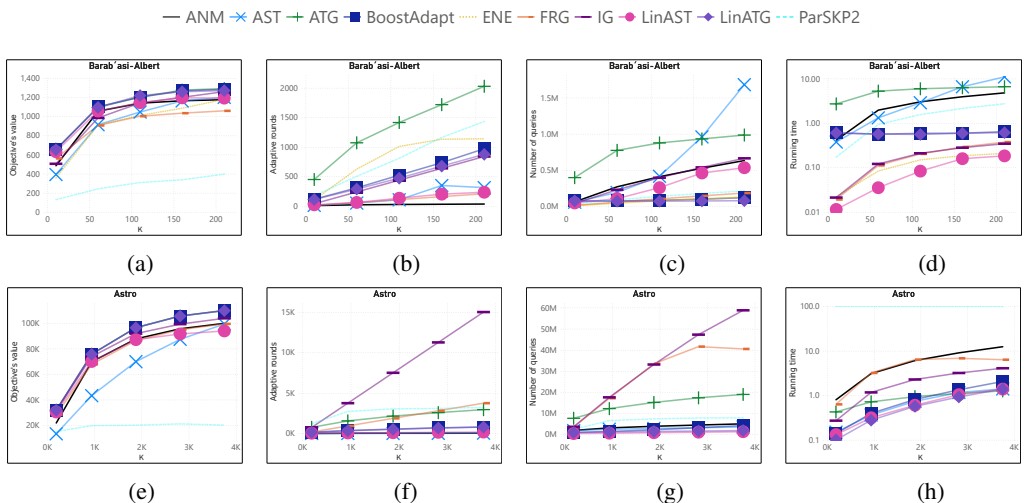

Figure 2: Performance of algorithms for Revenue Maximization (a-d) and Maximum Cut (e-h).

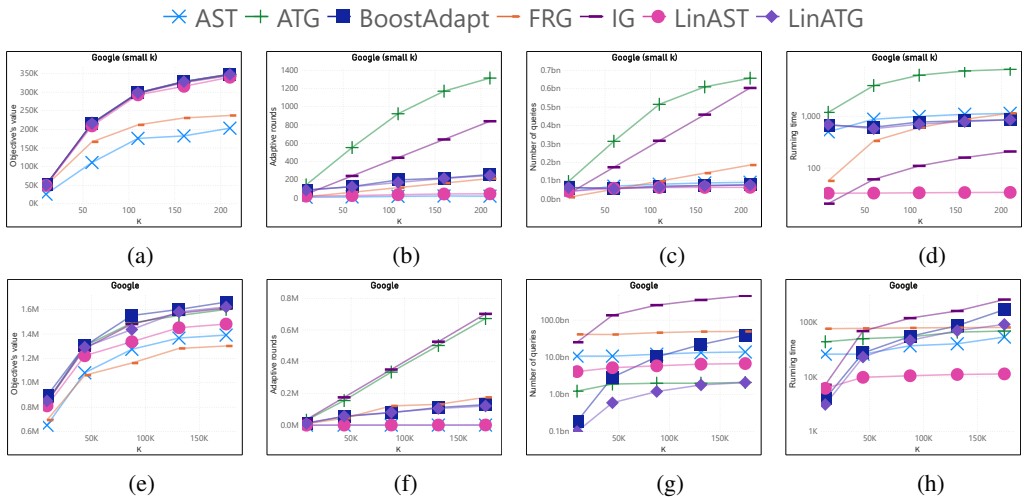

Figure 3: Performance of algorithms for Maximum Cut on large Google Dataset.

group of the lowest number of queries, followed by LinAst, ANM, ParSKP2 and IG algorithms. Meanwhile, AST and ATG require many queries. Nevertheless, in MC (Figure 2(g)), IG along with FRG reach highest points, followed by ATG. The others are substantially low.

The change also happens with the experiment of Google. With small $k$, ATG and IG waste much higher than the others. Our algorithms always mark the lowest points. With bigger $k$, LinAtg still hit the lowest points. LinAst, ATG and AST are close to LinAtg while BoostAdapt grows when $k$ grows. With high values of $k$, BoostAdapt reaches to FRG. IG wastes the highest number of queries. Finally, LinAtg and LinAst always keep steadily low while BoostAdapt keep low values in most cases except Google with large $k$.

**Time taken:** In the RM (Figure 2(d)), LinAst has the lowest execution time, followed by the ENE, IG and FRG algorithms. Our algorithms BoostAdapt and LinAtg have the same running time, which is higher than IG and FRG but lower than the remaining. In MC (Figure 2(h)), all our algorithms have the lowest points, a little lower than ATG and IG. ANM, ENE and FRG are moderately higher than the above algorithms while ParSKP2 has essentially highest values.

On the Google dataset (Figure 3), LinAst always runs fastest while others fluctuate. With small $k$, our BoostAdapt and LinAtg are almost equal to AST, higher than FRG and IG, and lower than ATG. With bigger $k$, IG runs slower than BoostAdapt and LinAtg and LinAst and have the trend run faster than the remaining.

The above metrics show that our algorithms outperform the others when keeping the best quality solutions, wasting the lowest the number of queries within acceptable query adaptivity.

## I ADDITIONAL LITERATURE REVIEW MONOTONE SMC

People studied non-adaptive methods for SMC first. Nemhauser et al. (1978) showed the best approximation algorithm with a factor of $1 - 1/e$ based on the sequential greedy approach. However, the sequential searching of greedy made time running too slow if the input increased. It led other works to try to reduce the time consumption problem. In Badanidiyuru & Vondrák (2014), they proposed a deterministic approximation algorithm to reduce time consumption down to $O(n \log n)$ by streaming fashion model. Significantly, Kuhnle (2021a) and Li et al. (2022) have simultaneously proposed a linear-time approximation algorithm with a tight factor of nearly $1 - 1/e$ recently. However, these algorithms cannot work for non-monotone functions.

To work with the approach of parallel, the first major contribution belonged to Balkanski & Singer (2018) when first applied the adaptive sampling method for the monotone SMC problem with constant 3-approximation ratio and $O(\log n)$ adaptivity. Their algorithm filtered a fixed fraction of the elements out of the ground set to get a quadratic query complexity. In contrast, Breuer et al. (2020) sped up the sampling method to enhance the approximation factor to $1 - 1/e$ within $O(\log n \log k)$ adaptive rounds and $O(n \log(\log k))$ queries.

