# OpenReview forum: "Boosting Parallel Algorithms in Linear Queries for Non-Monotone Submodular Maximization"
_ICLR.cc/2025/Conference — ICLR 2025 Conference Withdrawn Submission_

### Official Review · Reviewer_Fyjq · 2024-10-20

**Soundness:** 2
**Presentation:** 2
**Contribution:** 2
**Rating:** 6
**Confidence:** 4

**Summary:**

The paper presents three parallel algorithms, LinAst, LinAtg and BoostAdapt, for the problem of maximizing non-monotone submodular functions under cardinality constraints. These algorithms improve the approximation ratios and reduce query complexity over existing methods. The algorithms were tested on benchmarks related to applications like revenue maximization and maximum cut problems. The experimental results show the superiority of the proposed methods in terms of solution quality, query efficiency, and running time, compared to state-of-the-art algorithms.

**Strengths:**

The key strength of this paper lies in the introduction of two improved parallel algorithms, LinAst and LinAtg, as well as a new algorithm called BoostAdapt, for non-monotone submodular maximization. These algorithms enhance the approximation ratios while reducing query complexity from $O(n \log(k))$ to $O(n)$, and maintaining low adaptive complexity, making them suitable for large-scale parallel processing. BoostAdapt introduces a staggered greedy threshold framework, improving performance further. The experimental results validate the algorithms' superiority in real-world applications such as revenue maximization and maximum cut, demonstrating both theoretical rigor and practical effectiveness for big data scenarios.

**Weaknesses:**

The query complexity in (Amanatidis et al., 2021) and (Cui et al., 2023) can be reduced to near-linear using binary search techniques. While this does not yield results superior to those presented in this paper, it is important for the authors to provide an accurate account of prior work (especially in Table 1), allowing readers to better assess the contributions of this paper.

Moreover, the paper utilizes several results from existing literature. For example, many of the techniques and proof ideas in the key LinAdapt algorithm are derived from (Chen et al., 2021). Additionally, LinAst and LinAtg are essentially the AST and ATG algorithms proposed in (Chen & Kuhnle, 2024), indicating that the technical and theoretical contributions of this work are relatively limited. However, from my personal perspective, the contribution of combining existing techniques to achieve improved results is adequate, though not particularly impressive.

**Questions:**

Please refer to weaknesses.

---

### Official Review · Reviewer_Bgv1 · 2024-11-02

**Soundness:** 1
**Presentation:** 2
**Contribution:** 1
**Rating:** 3
**Confidence:** 5

**Summary:**

In this paper, the authors propose the first constant approximation algorithm, LinAdapt, with $O(\log n)$ adaptivity and $O(n)$ query complexity. Based on this result, they improve upon the algorithms in Chen & Kuhnle (2024), achieving $O(\log n)$ adaptivity and $O(n)$ query complexity while maintaining the same approximation ratios of $1/6-\epsilon$ and $0.193-\epsilon$. They further boost the approximation ratio to $1/4-\epsilon$ with $O(\log n \log k)$ adaptivity and $O(n\log k)$ query complexity. Experimental results show that their algorithms performs well in practice.

**Strengths:**

The paper improves the adaptivity and query complexity of algorithms in Chen & Kuhnle (2024) by reducing a factor of $O(\log k)$.

**Weaknesses:**

$\textbf{Limited Contribution}$:
1. The main subroutine of Alg. 1, LinBoundSet, is an adaptation of LinearSeq [1] and ThreshSeq [2], and its analysis closely follows [1] and [2]. However, the paper neither mentions these references nor clarifies their differences. Moreover, there is a mistake in the analysis of success probability, which is discussed in questions.
2. LinAst and LinAtg are essentially the same as AST and ATG in [2], with the only difference being the use of LinAdapt's output as input. The technical contribution is limited.
3. BoostAdapt is a novel algorithm; however, the analysis appears to contain errors. Please refer to questions.

$\textbf{The analysis in the paper is not rigorous.}$ As discussed above, there appear to be some errors, and the analysis is not detailed enough to verify its correctness. For example, Lemmata F.1, F.2, F.3, which are used to prove the approximation ratio of LinAtg are taken from [2] without being proved in this paper. Since the algorithms have different initial $\tau$ value, these lemmata require proof specific to this context. This lack of rigor makes it difficult to confirm the accuracy of the analysis.

$\textbf{Imprecise Expression}$:
1. Line 056: "Chen & Kuhnle (2024) demonstrated that Ene & Nguyen (2020)’s algorithm is of mostly theoretical interest because it needs $\Omega(nk^2 \log^2 (n))$ queries to access the multi-linear extension of a submodular function and its gradient." The multilinear extension and its gradient cannot be accessed, so "approximate" would be more accurate here.
2. Line 060: "The solution qualities of the existing ... are still low..." Does "solution qualities" refer to the objective values achieved in the experiments? As illustrated in previous works [3,2], FRG [4] with an approximation ratio of $1/e-\epsilon$ provides similar solution values to ANM [3], which has a $0.039-\epsilon$ approximation ratio, and ATG [2], which has a $0.193-\epsilon$ approximation ratio. This suggests that the challenge may lie in further improving the approximation ratio.
3. Line 095: "Therefore, our algorithm LinAtg improves the approximation ratio from $1/6 − \epsilon$ to $0.193 − \epsilon$..." ATG in [2] achieves $0.193-\epsilon$ approximation ratio. LinAtg improves on this by reducing the adaptivity and query complexity by a factor of $O(\log k)$.
4. Line 134: "To the best of our knowledge, the best ratio for SMC was 0.385 by (Buchbinder & Feldman, 2019)." The best ratio should be 0.401 by [5] as mentioned previously on Line 063.

$\textbf{Presentation Problem}$: The paper is well-structured and easy to follow. However, it contains numerous typos and grammatical errors, including but not limited to the following:
1. Line 099: "... guesses of the optimal." $\Rightarrow$ "... guesses of the optimal value."
2. Line 142: "... that measuring..." $\Rightarrow$ "... that measures..."
3. Line 142: "... a lower bound adaptive complexity..." $\Rightarrow$ "... a lower bound on adaptive complexity..."
4. Line 153: "However, due to the high query complexity..." It is not a complete sentence."
5. Line 171: "... $f: 2^V \to \mathbb{R}^+$ is submodularity." $\Rightarrow$ "... $f: 2^V \to \mathbb{R}^+$ is submodular."
6. Table 1: It is said that the best results are bold. So, $1/e-\epsilon$, $O(\log n)$, and $O(n)$ should be bold under the three columns. Instead, only the results in this work are bold.

References:

[1] Yixin Chen, Tonmoy Dey, and Alan Kuhnle. Best of both worlds: Practical and theoretically optimal submodular maximization in parallel. In Marc’Aurelio Ranzato, Alina Beygelzimer, Yann N. Dauphin, Percy Liang, and Jennifer Wortman Vaughan (eds.), Advances in Neural Information Processing Systems 34: Annual Conference on Neural Information Processing Systems 2021, NeurIPS 2021, December 6-14, 2021, virtual, pp. 25528–25539, 2021.

[2] Yixin Chen and Alan Kuhnle. Practical and parallelizable algorithms for non-monotone submodular maximization with size constraint. Journal of Artificial Intelligence Research, 79:599–637, 2024.

[3] Matthew Fahrbach, Vahab Mirrokni, and Morteza Zadimoghaddam. Non-monotone submodular maximization with nearly optimal adaptivity and query complexity. preprint, arXiv:1808.06932, 2023. URL https://arxiv.org/abs/1808.06932.

[4] Niv Buchbinder, Moran Feldman, and Roy Schwartz. Comparing apples and oranges: Query tradeoff in submodular maximization. In Proc. of the 26thl ACM-SIAM SODA 2015, pp. 1149–1168, 2015.

[5] Niv Buchbinder and Moran Feldman. Constrained submodular maximization via new bounds for dr-submodular functions. In Proceedings of the 56th Annual ACM Symposium on Theory of Computing, STOC 2024, Vancouver, BC, Canada, June 24-28, 2024, pp. 1820–1831. ACM, 2024.

**Questions:**

$\textbf{Possible Incorrect Result}$
1. The analysis of success probability of Alg. 2 is wrong. On Page 22, Line 1157, it states "The probability iteration $j$ failure is upper bounded by the probability $\lambda^* < t$." By the definitions in Proposition E.1 and E.2 and the definition of the failure of iteration $j$, it seems that $Q_{\lambda^*}$ is considered to be the elements that will be deleted in the next iteration. However, the set $S$ used to update $V$ on Line 4 in Alg. 2 is not the same as $S\cup T_t$ in Proposition E.1. Specifically, on Lines 20-21 in Alg. 2, a subset of $T_{\lambda^*}$ is added to S.
2. The analysis of approximation ratio of Alg. 5 seems wrong. In Inequality (152) on Page 29, why $f(e|X) < \tau_X^{last}$, for all $e \in O\setminus (X\cup Y^+)$? Elements in $O\setminus (X\cup Y^+)$ may still be in $Y$, where we can not get $f(e|X) < \tau_X^{last}$.

$\textbf{Question on Experiments}$

The paper provides empirical results on four datasets, which are the same in [1]. However, the results are very different. For example, in [1], ATG achieves the highest objective value on the astro dataset, followed by AST and FRG, with ANM performing the worst. In contrast, in this paper, AST performs significantly worse than the other methods on the same dataset. Same as the other datasets.

Additionally, consider the differences between AST in [1] and LinAst. The for loop in AST starts with $\tau$ equals to the maximum singleton value $M$, and ends with $\tau\le M/(ck)$. In LinAst, the for loop starts with $\tau = af(S_0)/(ck)$, and ends with $\tau \le f(S_0)/(ck)$. If $af(S_0)/(ck) > M$, ThreshSeq will return empty set until $\tau \le M$. Therefore, the largest $\tau$ value LinAst tried is less than or equal to AST. As for the smallest $\tau$ values, if $f(S_0) \ge M$, then the smallest $\tau$ value LinAst tried is larger than or equal to AST. Thus, the $\tau$ values tried by LinAst should be included within those tried by AST (since the starting values are different, it won't be exactly the same but would be close). Moreover, the objective value of AST should not be significantly worse than that of LinAst. If $f(S_0) < M$, the max singleton value would be quite large since $f(S_0)$ is derived from a deterministic $1/12$-approximation algorithm. It seems unlikely that this scenario would occur across all the datasets. In summary, I do not think the objective value returned by AST and LinAst would have much difference.

References:

[1] Yixin Chen and Alan Kuhnle. Practical and parallelizable algorithms for non-monotone submodular maximization with size constraint. Journal of Artificial Intelligence Research, 79:599–637, 2024.

---

### Official Review · Reviewer_Wfek · 2024-11-03

**Soundness:** 3
**Presentation:** 3
**Contribution:** 3
**Rating:** 5
**Confidence:** 4

**Summary:**

This work proposes new low-adaptivity algorithms for cardinality-constrained non-monotone submodular maximization
while also taking into account the practical constraints of low query complexity.
Specifically, it proposes the `LinAtg` algorithm with:
  * Adaptivity complexity: $O(\log n)$
  * Approximation ratio: $0.193 - \varepsilon$
  * Query complexity $O(n)$

as well as an enhanced `BoostAdapt` algorithm with:
  * Adaptivity complexity: $O(\log n \log k)$
  * Approximation ratio: $0.25 - \varepsilon$
  * Query complexity: $O(n \log k)$

For context (see Table 1), this improves on the recent work of [Chen-Kuhnle, JAIR 2024] that achieves
a $(0.193-\varepsilon)$-approximation ratio in $O(\log n)$ adaptive rounds with $O(n \log k)$ queries,
and also the work of [Cui et al., AAAI 2023] that achieves a $(0.25-\varepsilon)$-approximation ratio
in $O(\log n \log k)$ adaptive rounds with $O(nk \log^2 n)$ queries.

Overall, the main ideas are well organized and easy to follow, especially in
the context of previous work which the authors do a good job of connecting to.
Finally, the authors provide a sufficient set of experiments that empirically
compare their new algorithms to previous methods on revenue maximization and maximum cut tasks.

**Strengths:**

- Designs low-adaptivity algorithms with query complexity in mind (which can be
  prohibitively expensive, even in parallel settings)
- Comprehensive study of low-adaptibity algorithms (Table 1), as well as best-known approximatino algorithm (0.401 in [Buchbinder-Feldman, STOC 2024])
- `LinAst` and `BoostAdapt` both clearly push the {approximation ratio, adaptivity, query complexity} Pareto frontier
- Uses OpenMP for parallelization, which means the wall time metrics are more realistic
- `BoostAdapt uses a novel staggered greedy technique that alternatately
  constructs two disjoint sets, which differs from Iterated Greedy, twin greedy, and interlaced greedy ideas.

**Weaknesses:**

- This isn't necessarily a weakness, but there are a lot of ideas going on
  which makes it hard to fit everything into the page limit. In this case, it
  seems like the empirical study got the short end of the stick. Some examples:
  * It would be beneficial to include more plots in the main body to paint a fuller picture. One dataset per tasks is insufficient, though they are given in Appendix H.
  * The revenue maximization and max cut tasks are well studied (plus can be related to other works), but this work could also benefit from a new set of experiments.
  * Subplots should also have the tasks in the titles so that we don't need to read the caption to index the plots.
- It is unclear if ICLR is an appropriate venue for this work. The
  non-exhaustive list of topics in the Call for Papers includes "optimization",
  but submodular maximization in its raw form seems one hop away from the target areas of ICLR (deep learning).
- The writing could be tightened up (e.g., "saves the best adaptive rounds" --> "uses the fewest adaptive rounds")

**Questions:**

- In Section 2 you write "To the best of our knowledge, the best ratio for SMC was $0.385$ by [Buchbinder-Filedman, 2019]," but you have also written
  "... the best non-parallel algorithm for SMC (e.g., the raito of $0.401$ in [Buchbinder-Feldman, 2024])." Which is it?
- In Section 7, why do you choose to set $\alpha = 4$? In Theorem 4.1, you say
  that "if $\alpha = 1$, the algorithm achieves the best ratio of $1/12$." If this
  is the result of a hyperparameter sweep, it is best to make that clear.
- Why are not all algorithms present in Figure 3 (Appendix H)?

**Misc**
- Missing space before "(ParSKP1)" in Table 1
- On page 5, "running in $O(\log (n) / \epsilon^2)$ and" --> "running in $O(\log (n) / \epsilon^2)$ adaptive rounds and"

---

### Official Review · Reviewer_bf7c · 2024-11-08

**Soundness:** 3
**Presentation:** 4
**Contribution:** 2
**Rating:** 6
**Confidence:** 3

**Summary:**

This paper studies the problem of maximizing a non-monotone submodular function under a cardinality constraint. The authors propose new parallel algorithms that aim to improve approximation ratios and query complexity while maintaining low adaptive complexity. The authors provide both theoretical analysis and experimental results, indicating their algorithms offer advantages in practical applications by improving state-of-the-art performance across multiple metrics.

**Strengths:**

The authors present algorithms that improve at least one metric—approximation factor, adaptive complexity, or query complexity—while holding other parameters steady. For example, LinAst and LinAtg reduce query complexity without increasing adaptive complexity, achieving balanced improvements over existing approaches.
The paper provides experimental evidence demonstrating that the proposed algorithms outperform previous methods in solution quality, adaptive rounds, and runtime. This practical perspective supports the algorithms’ applicability in real-world settings.
The authors detail their theoretical guarantees, providing sound proofs that make the contributions accessible and relevant to both theoretical and applied researchers in submodular maximization.

**Weaknesses:**

When considering each metric—approximation factor, adaptivity, and query complexity—individually, this work does not improve upon the best-known value for any single metric. For instance, the best approximation factor remains at 1/4; however, they have achieved a lower query complexity for an algorithm with this approximation factor.

Minor comment:
In the abstract, it is mentioned that for algorithms with an adaptive complexity of O(log⁡(n)), you have improved the approximation ratio from 1/6 to 0.193. However, it seems that before your work, the best approximation was 0.172 by Amanatidis et al. [13], not 1/6.

**Questions:**

None

---

### Note · Authors · 2024-11-26

I have read and agree with the venue's withdrawal policy on behalf of myself and my co-authors.